# AN ADAPTIVE POLICY TO EMPLOY SHARPNESS-AWARE MINIMIZATION

**Weisen Jiang[1, 2], Hansi Yang[2], Yu Zhang[1, 3, *], James Kwok[2]**

[1] Guangdong Provincial Key Laboratory of Brain-inspired Intelligent Computation
  Department of Computer Science and Engineering, Southern University of Science and Technology
[2] Department of Computer Science and Engineering, Hong Kong University of Science and Technology
[3] Peng Cheng Laboratory
{wjiangar, hyangbw, jamesk}@cse.ust.hk, yu.zhang.ust@gmail.com

## ABSTRACT

Sharpness-aware minimization (SAM), which searches for flat minima by min-max optimization, has been shown to be useful in improving model generalization. However, since each SAM update requires computing two gradients, its computational cost and training time are both doubled compared to standard empirical risk minimization (ERM). Recent state-of-the-arts reduce the fraction of SAM updates and thus accelerate SAM by switching between SAM and ERM updates randomly or periodically. In this paper, we design an adaptive policy to employ SAM based on the loss landscape geometry. Two efficient algorithms, AE-SAM and AE-LookSAM, are proposed. We theoretically show that AE-SAM has the same convergence rate as SAM. Experimental results on various datasets and architectures demonstrate the efficiency and effectiveness of the adaptive policy.

## 1 INTRODUCTION

Despite great success in many applications (He et al., 2016; Zagoruyko & Komodakis, 2016; Han et al., 2017), deep networks are often over-parameterized and capable of memorizing all training data. The training loss landscape is complex and nonconvex with many local minima of different generalization abilities. Many studies have investigated the relationship between the loss surface's geometry and generalization performance (Hochreiter & Schmidhuber, 1994; McAllester, 1999; Keskar et al., 2017; Neyshabur et al., 2017; Jiang et al., 2020), and found that flatter minima generalize better than sharper minima (Dziugaite & Roy, 2017; Petzka et al., 2021; Chaudhari et al., 2017; Keskar et al., 2017; Jiang et al., 2020).

Sharpness-aware minimization (SAM) (Foret et al., 2021) is the current state-of-the-art to seek flat minima by solving a min-max optimization problem. In the SAM algorithm, each update consists of *two* forward-backward computations: one for computing the perturbation and the other for computing the actual update direction. Since these two computations are not parallelizable, SAM doubles the computational overhead as well as the training time compared to empirical risk minimization (ERM).

Several algorithms (Du et al., 2022a; Zhao et al., 2022b; Liu et al., 2022) have been proposed to improve the efficiency of SAM. ESAM (Du et al., 2022a) uses fewer samples to compute the gradients and updates fewer parameters, but each update still requires two gradient computations. Thus, ESAM does not alleviate the bottleneck of training speed. Instead of using the SAM update at every iteration, recent state-of-the-arts (Zhao et al., 2022b; Liu et al., 2022) proposed to use SAM randomly or periodically. Specifically, SS-SAM (Zhao et al., 2022b) selects SAM or ERM according to a Bernoulli trial, while LookSAM (Liu et al., 2022) employs SAM at every $k$ step. Though more efficient, the random or periodic use of SAM is suboptimal as it is not geometry-aware. Intuitively, the SAM update is more useful in sharp regions than in flat regions.

In this paper, we propose an adaptive policy to employ SAM based on the geometry of the loss landscape. The SAM update is used when the model is in sharp regions, while the ERM update is used in flat regions for reducing the fraction of SAM updates. To measure sharpness, we use

---

*Correspondence to: Yu Zhang

the squared stochastic gradient norm and model it by a normal distribution, whose parameters are estimated by exponential moving average. Experimental results on standard benchmark datasets demonstrate the superiority of the proposed policy.

Our contributions are summarized as follows: (i) We propose an adaptive policy to use SAM or ERM update based on the loss landscape geometry. (ii) We propose an efficient algorithm, called AE-SAM (Adaptive policy to Employ SAM), to reduce the fraction of SAM updates. We also theoretically study its convergence rate. (iii) The proposed policy is general and can be combined with any SAM variant. In this paper, we integrate it with LookSAM (Liu et al., 2022) and propose AE-LookSAM. (iv) Experimental results on various network architectures and datasets (with and without label noise) verify the superiority of AE-SAM and AE-LookSAM over existing baselines.

**Notations**. Vectors (e.g., $\mathbf{x}$) and matrices (e.g., $\mathbf{X}$) are denoted by lowercase and uppercase boldface letters, respectively. For a vector $\mathbf{x}$, its $\ell_2$-norm is $\|\mathbf{x}\|$. $\mathcal{N}(\mu; \sigma^2)$ is the univariate normal distribution with mean $\mu$ and variance $\sigma^2$. $\mathrm{diag}(\mathbf{x})$ constructs a diagonal matrix with $\mathbf{x}$ on the diagonal. Moreover, $\mathbb{I}_A(x)$ denotes the indicator function for a given set $A$, i.e., $\mathbb{I}_A(x) = 1$ if $x \in A$, and 0 otherwise.

## 2 RELATED WORK

We are given a training set $\mathcal{D}$ with i.i.d. samples $\{(\mathbf{x}_i, y_i) : i = 1, \ldots, n\}$. Let $f(\mathbf{x}; \mathbf{w})$ be a model parameterized by $\mathbf{w}$. Its empirical risk on $\mathcal{D}$ is $\mathcal{L}(\mathcal{D}; \mathbf{w}) = \frac{1}{n} \sum_{i=1}^{n} \ell(f(\mathbf{x}_i; \mathbf{w}), y_i)$, where $\ell(\cdot, \cdot)$ is a loss (e.g., cross-entropy loss for classification). Model training aims to learn a model from the training data that generalizes well on the test data.

**Generalization and Flat Minima.** The connection between model generalization and loss landscape geometry has been theoretically and empirically studied in (Keskar et al., 2017; Dziugaite & Roy, 2017; Jiang et al., 2020). Recently, Jiang et al. (2020) conducted large-scale experiments and find that sharpness-based measures (flatness) are related to generalization of minimizers. Although flatness can be characterized by the Hessian's eigenvalues (Keskar et al., 2017; Dinh et al., 2017), handling the Hessian explicitly is computationally prohibitive. To address this issue, practical algorithms propose to seek flat minima by injecting noise into the optimizers (Zhu et al., 2019; Zhou et al., 2019; Orvieto et al., 2022; Bisla et al., 2022), introducing regularization (Chaudhari et al., 2017; Zhao et al., 2022a; Du et al., 2022b), averaging model weights during training (Izmailov et al., 2018; He et al., 2019; Cha et al., 2021), or sharpness-aware minimization (SAM) (Foret et al., 2021; Kwon et al., 2021; Zhuang et al., 2022; Kim et al., 2022).

**SAM.** The state-of-the-art SAM (Foret et al., 2021) and its variants (Kwon et al., 2021; Zhuang et al., 2022; Kim et al., 2022; Zhao et al., 2022a) search for flat minima by solving the following min-max optimization problem:

$$\min_{\mathbf{w}} \max_{\|\boldsymbol{\epsilon}\| \leq \rho} \mathcal{L}(\mathcal{D}; \mathbf{w} + \boldsymbol{\epsilon}), \tag{1}$$

where $\rho > 0$ is the radius of perturbation. The above can also be rewritten as $\min_{\mathbf{w}} \mathcal{L}(\mathcal{D}; \mathbf{w}) + \mathcal{R}(\mathcal{D}; \mathbf{w})$, where $\mathcal{R}(\mathcal{D}; \mathbf{w}) \equiv \max_{\|\boldsymbol{\epsilon}\| \leq \rho} \mathcal{L}(\mathcal{D}; \mathbf{w} + \boldsymbol{\epsilon}) - \mathcal{L}(\mathcal{D}; \mathbf{w})$ is a regularizer that penalizes sharp minimizers (Foret et al., 2021). As solving the inner maximization in (1) exactly is computationally infeasible for nonconvex losses, SAM approximately solves it by first-order Taylor approximation, leading to the update rule:

$$\mathbf{w}_{t+1} = \mathbf{w}_t - \eta \nabla \mathcal{L}(\mathcal{B}_t; \mathbf{w}_t + \rho_t \nabla \mathcal{L}(\mathcal{B}_t; \mathbf{w}_t)), \tag{2}$$

where $\mathcal{B}_t$ is a mini-batch of data, $\eta$ is the step size, and $\rho_t = \frac{\rho}{\|\nabla \mathcal{L}(\mathcal{B}_t; \mathbf{w}_t)\|}$. Although SAM has shown to be effective in improving the generalization of deep networks, a major drawback is that each update in (2) requires *two* forward-backward calculations. Specifically, SAM first calculates the gradient of $\mathcal{L}(\mathcal{B}_t; \mathbf{w})$ at $\mathbf{w}_t$ to obtain the perturbation, then calculates the gradient of $\mathcal{L}(\mathcal{B}_t; \mathbf{w})$ at $\mathbf{w}_t + \rho_t \nabla \mathcal{L}(\mathcal{B}_t; \mathbf{w}_t)$ to obtain the update direction for $\mathbf{w}_t$. As a result, SAM doubles the computational overhead compared to ERM.

**Efficient Variants of SAM.** Several algorithms have been proposed to accelerate the SAM algorithm. ESAM (Du et al., 2022a) uses fewer samples to compute the gradients and only updates part of the model in the second step, but still requires to compute most of the gradients. Another direction is to reduce the number of SAM updates during training. SS-SAM (Zhao et al., 2022b) randomly selects

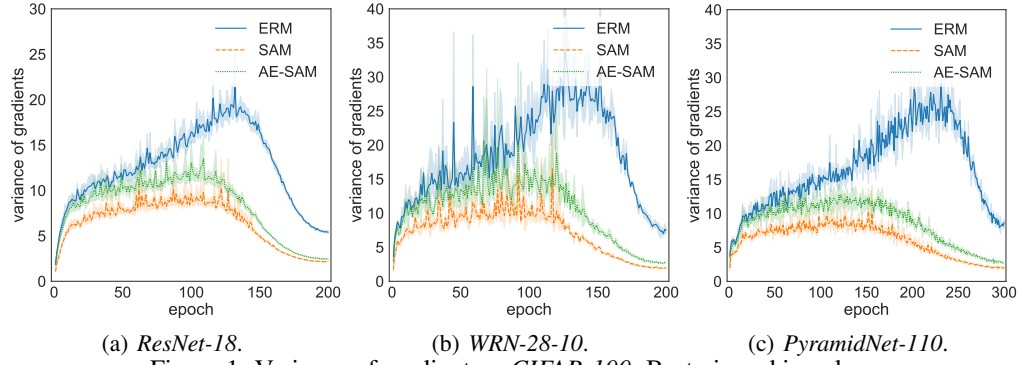

(a) *ResNet-18*.          (b) *WRN-28-10*.          (c) *PyramidNet-110*.

Figure 1: Variance of gradient on *CIFAR-100*. Best viewed in color.

SAM or ERM update according to a Bernoulli trial, while LookSAM (Liu et al., 2022) employs SAM at every $k$ iterations. Intuitively, the SAM update is more suitable for sharp regions than flat regions. However, the mixing policies in SS-SAM and LookSAM are not adaptive to the loss landscape. In this paper, we design an adaptive policy to employ SAM based on the loss landscape geometry.

## 3 METHOD

In this section, we propose an adaptive policy to employ SAM. The idea is to use ERM when $\mathbf{w}_t$ is in a flat region, and use SAM only when the loss landscape is locally sharp. We start by introducing a sharpness measure (Section 3.1), then propose an adaptive policy based on this (Section 3.2). Next, we propose two algorithms (AE-SAM and AE-LookSAM) and study the convergence.

### 3.1 SHARPNESS MEASURE

Though sharpness can be characterized by Hessian's eigenvalues (Keskar et al., 2017; Dinh et al., 2017), they are expensive to compute. A widely-used approximation is based on the gradient magnitude $\mathrm{diag}([\nabla\mathcal{L}(\mathcal{B}_t;\mathbf{w}_t)]^2)$ (Bottou et al., 2018; Khan et al., 2018), where $[\mathbf{v}]^2$ denotes the elementwise square of a vector $\mathbf{v}$. As $\|\nabla\mathcal{L}(\mathcal{B}_t;\mathbf{w}_t)\|^2$ equals the trace of $\mathrm{diag}([\nabla\mathcal{L}(\mathcal{B}_t;\mathbf{w}_t)]^2)$, it is reasonable to choose $\|\nabla\mathcal{L}(\mathcal{B}_t;\mathbf{w}_t)\|^2$ as a sharpness measure.

$\|\nabla\mathcal{L}(\mathcal{B}_t;\mathbf{w}_t)\|^2$ is also related to the gradient variance $\mathsf{Var}(\nabla\mathcal{L}(\mathcal{B}_t;\mathbf{w}_t))$, another sharpness measure (Jiang et al., 2020). Specifically,

$$\mathsf{Var}(\nabla\mathcal{L}(\mathcal{B}_t;\mathbf{w}_t)) \equiv \mathbb{E}_{\mathcal{B}_t}\|\nabla\mathcal{L}(\mathcal{B}_t;\mathbf{w}_t) - \nabla\mathcal{L}(\mathcal{D};\mathbf{w}_t)\|^2 = \mathbb{E}_{\mathcal{B}_t}\|\nabla\mathcal{L}(\mathcal{B}_t;\mathbf{w}_t)\|^2 - \|\nabla\mathcal{L}(\mathcal{D};\mathbf{w}_t)\|^2. \quad (3)$$

With appropriate smoothness assumptions on $\mathcal{L}$, both SAM and ERM can be shown theoretically to converge to critical points of $\mathcal{L}(\mathcal{D};\mathbf{w})$ (i.e., $\nabla\mathcal{L}(\mathcal{D};\mathbf{w}) = 0$) (Reddi et al., 2016; Andriushchenko & Flammarion, 2022). Thus, it follows from (3) that $\mathsf{Var}(\nabla\mathcal{L}(\mathcal{B}_t;\mathbf{w}_t)) = \mathbb{E}_{\mathcal{B}_t}\|\nabla\mathcal{L}(\mathcal{B}_t;\mathbf{w}_t)\|^2$ when $\mathbf{w}_t$ is a critical point of $\mathcal{L}(\mathcal{D};\mathbf{w})$. Jiang et al. (2020) conducted extensive experiments and empirically show that $\mathsf{Var}(\nabla\mathcal{L}(\mathcal{B}_t;\mathbf{w}_t))$ is positively correlated with the generalization gap. The smaller the $\mathsf{Var}(\nabla\mathcal{L}(\mathcal{B}_t;\mathbf{w}_t))$, the better generalization is the model with parameter $\mathbf{w}_t$. This finding also explains why SAM generalizes better than ERM. Figure 1 shows the gradient variance w.r.t. the number of epochs using SAM and ERM on *CIFAR-100* with various network architectures (experimental details are in Section 4.1). As can be seen, SAM always has a much smaller variance than ERM. Figure 2 shows the expected squared norm of the stochastic gradient w.r.t. the number of epochs on *CIFAR-100*. As shown, SAM achieves a much smaller $\mathbb{E}_{\mathcal{B}_t}\|\nabla\mathcal{L}(\mathcal{B}_t;\mathbf{w}_t)\|^2$ than ERM.

### 3.2 ADAPTIVE POLICY TO EMPLOY SAM

As $\mathbb{E}_{\mathcal{B}_t}\|\nabla\mathcal{L}(\mathcal{B}_t;\mathbf{w}_t)\|^2$ changes with $t$ (Figure 2), the sharpness at $\mathbf{w}_t$ also changes along the optimization trajectory. As a result, we need to estimate $\mathbb{E}_{\mathcal{B}_t}\|\nabla\mathcal{L}(\mathcal{B}_t;\mathbf{w}_t)\|^2$ at every iteration. One can sample a large number of mini-batches and compute the mean of the stochastic gradient norms. However, this can be computationally expensive. To address this problem, we model $\|\nabla\mathcal{L}(\mathcal{B}_t;\mathbf{w}_t)\|^2$ with a simple distribution and estimate the distribution parameters in an online manner. Figure 3(a) shows $\|\nabla\mathcal{L}(\mathcal{B}_t;\mathbf{w}_t)\|^2$ of 400 mini-batches at different training stages (epoch = 60, 120, and 180)

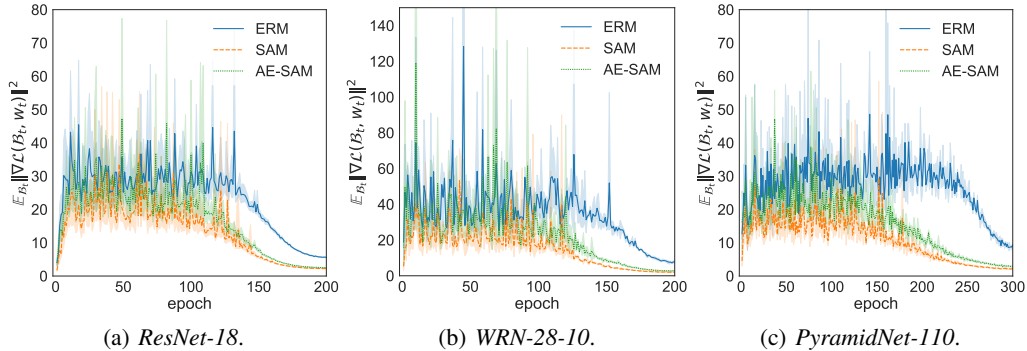

(a) *ResNet-18.*  (b) *WRN-28-10.*  (c) *PyramidNet-110.*

Figure 2: Squared stochastic gradient norms $\mathbb{E}_{\mathcal{B}}\|\nabla\mathcal{L}(\mathcal{B};\mathbf{w}_t)\|^2$ on *CIFAR-100*. Best viewed in color.

on *CIFAR-100* using *ResNet-18*[1]. As can be seen, the distribution follows a Bell curve. Figure 3(b) shows the corresponding quantile-quantile (Q-Q) plot (Wilk & Gnanadesikan, 1968). The closer is the curve to a line, the distribution is closer to the normal distribution. Figure 3 suggests that $\|\nabla\mathcal{L}(\mathcal{B}_t;\mathbf{w}_t)\|^2$ can be modeled[2] with a normal distribution $\mathcal{N}(\mu_t, \sigma_t^2)$. We use exponential moving average (EMA), which is popularly used in adaptive gradient methods (e.g., RMSProp (Tieleman & Hinton, 2012), AdaDelta (Zeiler, 2012), Adam (Kingma & Ba, 2015)), to estimate its mean and variance:

$$\mu_t = \delta\mu_{t-1} + (1-\delta)\|\nabla\mathcal{L}(\mathcal{B}_t;\mathbf{w}_t)\|^2, \tag{4}$$

$$\sigma_t^2 = \delta\sigma_{t-1}^2 + (1-\delta)(\|\nabla\mathcal{L}(\mathcal{B}_t;\mathbf{w}_t)\|^2 - \mu_t)^2, \tag{5}$$

where $\delta \in (0,1)$ controls the forgetting rate. Empirically, we use $\delta = 0.9$. Since $\nabla\mathcal{L}(\mathcal{B}_t;\mathbf{w}_t)$ is already available during training, this EMA update does not involve additional gradient calculations (the cost for the norm operator is negligible).

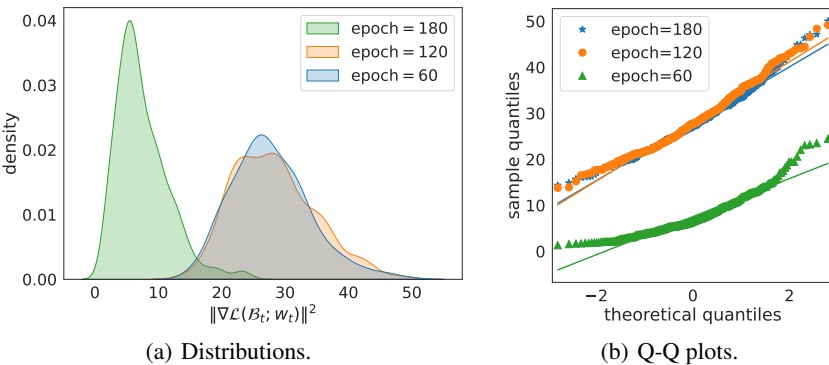

(a) Distributions.  (b) Q-Q plots.

Figure 3: Stochastic gradient norms $\{\|\mathcal{L}(\mathcal{B}_t;\mathbf{w}_t)\|^2 : \mathcal{B}_t \sim \mathcal{D}\}$ of *ResNet-18* on *CIFAR-100* are approximately normally distributed. Best viewed in color.

Using $\mu_t$ and $\sigma_t^2$, we employ SAM only at iterations where $\|\nabla\mathcal{L}(\mathcal{B}_t;\mathbf{w}_t)\|^2$ is relatively large (i.e., the loss landscape is locally sharp). Specifically, when $\|\nabla\mathcal{L}(\mathcal{B}_t;\mathbf{w}_t)\|^2 \geq \mu_t + c_t\sigma_t$ (where $c_t$ is a threshold), SAM is used; otherwise, ERM is used. When $c_t \to -\infty$, it reduces to SAM; when $c_t \to \infty$, it becomes ERM. Note that during the early training stage, the model is still underfitting and $\mathbf{w}_t$ is far from the region of final convergence. Thus, minimizing the empirical loss is more important than seeking a locally flat region. Andriushchenko & Flammarion (2022) also empirically observe that the SAM update is more effective in boosting performance towards the end of training. We therefore design a schedule that linearly decreases $c_t$ from $\lambda_2$ to $\lambda_1$ (which are pre-set values): $c_t = g_{\lambda_1,\lambda_2}(t) \equiv \frac{t}{T}\lambda_1 + \left(1 - \frac{t}{T}\right)\lambda_2$, where $T$ is the total number of iterations. The whole procedure, called $\underline{A}$daptive policy to $\underline{E}$mploy $\underline{SAM}$ (**AE-SAM**), is shown in Algorithm 1.

**AE-LookSAM**. The proposed adaptive policy can be combined with any SAM variant. Here, we consider integrating it with LookSAM (Liu et al., 2022). When $\|\nabla\mathcal{L}(\mathcal{B}_t;\mathbf{w}_t)\|^2 \geq \mu_t + c_t\sigma_t$, SAM

---

[1]Results on other architectures and *CIFAR-10* are shown in Figures 8 and 9 of Appendix B.1.
[2]Note that normality is not needed in the theoretical analysis (Section 3.3).

is used and the update direction for $\mathbf{w}_t$ is decomposed into two orthogonal directions as in LookSAM: (i) the ERM update direction to reduce training loss, and (ii) the direction that biases the model to a flat region. When $\|\nabla\mathcal{L}(\mathcal{B}_t; \mathbf{w}_t)\|^2 < \mu_t + c_t\sigma_t$, ERM is performed and the second direction of the previous SAM update is reused to compose an approximate SAM direction. The procedure, called AE-LookSAM, is also shown in Algorithm 1.

---

**Algorithm 1** AE-SAM and AE-LookSAM.

---

**Require:** training set $\mathcal{D}$, stepsize $\eta$, radius $\rho$; $\lambda_1$ and $\lambda_2$ for $g_{\lambda_1,\lambda_2}(t)$; $\mathbf{w}_0$, $\mu_{-1} = 0$, $\sigma_{-1}^2 = e^{-10}$, and $\alpha$ for AE-LookSAM;
 1: **for** $t = 0, \ldots, T-1$ **do**
 2:     sample a mini-batch data $\mathcal{B}_t$ from $\mathcal{D}$;
 3:     compute $\mathbf{g} = \nabla\mathcal{L}(\mathcal{B}_t; \mathbf{w}_t)$;
 4:     update $\mu_t$ by (4) and $\sigma_t^2$ by (5);
 5:     compute $c_t = g_{\lambda_1,\lambda_2}(t)$;
 6:     **if** $\|\nabla\mathcal{L}(\mathcal{B}_t; \mathbf{w}_t)\|^2 \geq \mu_t + c_t\sigma_t$ **then**
 7:         $\mathbf{g}_s = \nabla\mathcal{L}(\mathcal{B}_t; \mathbf{w}_t + \rho\nabla\mathcal{L}(\mathcal{B}_t; \mathbf{w}_t))$;
 8:         if AE-LookSAM: decompose $\mathbf{g}_s$ as $\mathbf{g}_v = \mathbf{g}_s - \frac{\mathbf{g}^\top \mathbf{g}_s}{\|\mathbf{g}\|^2}\mathbf{g}$;
 9:     **else**:
10:         if AE-SAM: $\mathbf{g}_s = \mathbf{g}$;
11:         if AE-LookSAM: $\mathbf{g}_s = \mathbf{g} + \alpha\frac{\|\mathbf{g}\|}{\|\mathbf{g}_v\|}\mathbf{g}_v$;
12:     **end if**
13:     $\mathbf{w}_{t+1} = \mathbf{w}_t - \eta\mathbf{g}_s$;
14: **end for**
15: **return** $\mathbf{w}_T$.

---

### 3.3 CONVERGENCE ANALYSIS

In this section, we study the convergence of any algorithm $\mathcal{A}$ whose update in each iteration can be either SAM or ERM. Due to this mixing of SAM and ERM updates, analyzing its convergence is more challenging compared with that of SAM.

The following assumptions on smoothness and bounded variance of stochastic gradients are standard in the literature on non-convex optimization (Ghadimi & Lan, 2013; Reddi et al., 2016) and SAM (Andriushchenko & Flammarion, 2022; Abbas et al., 2022; Qu et al., 2022).

**Assumption 3.1** (Smoothness). $\mathcal{L}(\mathcal{D}; \mathbf{w})$ is $\beta$-smooth in $\mathbf{w}$, i.e., $\|\nabla\mathcal{L}(\mathcal{D}; \mathbf{w}) - \nabla\mathcal{L}(\mathcal{D}; \mathbf{v})\| \leq \beta\|\mathbf{w} - \mathbf{v}\|$.

**Assumption 3.2** (Bounded variance of stochastic gradients). $\mathbb{E}_{(\mathbf{x}_i, y_i)\sim\mathcal{D}}\|\nabla\ell(f(\mathbf{x}_i; \mathbf{w}), y_i) - \nabla\mathcal{L}(\mathcal{D}; \mathbf{w})\|^2 \leq \sigma^2$.

Let $\xi_t$ be an indicator of whether SAM or ERM is used at iteration $t$ (i.e., $\xi_t = 1$ for SAM, and 0 for ERM). For example, $\xi_t = \mathbb{I}_{\{\mathbf{w}:\|\nabla\mathcal{L}(\mathcal{B}_t;\mathbf{w})\|^2 \geq \mu_t + c_t\sigma_t\}}(\mathbf{w}_t)$ for the proposed AE-SAM, and $\xi_t$ is sampled from a Bernoulli distribution for SS-SAM (Zhao et al., 2022b).

**Theorem 3.3.** *Let $b$ be the mini-batch size. If stepsize $\eta = \frac{1}{4\beta\sqrt{T}}$ and $\rho = \frac{1}{T^{\frac{1}{4}}}$, algorithm $\mathcal{A}$ satisfies*

$$\min_{0\leq t\leq T-1}\mathbb{E}\|\nabla\mathcal{L}(\mathcal{D}; \mathbf{w}_t)\|^2 \leq \frac{32\beta\left(\mathcal{L}(\mathcal{D}; \mathbf{w}_0) - \mathbb{E}\mathcal{L}(\mathcal{D}; \mathbf{w}_T)\right)}{\sqrt{T}\,(7 - 6\zeta)} + \frac{(1 + \zeta + 5\beta^2\zeta)\sigma^2}{b\sqrt{T}\,(7 - 6\zeta)}, \quad (6)$$

*where $\zeta = \frac{1}{T}\sum_{t=0}^{T-1}\xi_t \in [0, 1]$ is the fraction of SAM updates, and the expectation is taken over the random training samples.*

All proofs are in Appendix A. Note that a larger $\zeta$ leads to a larger upper bound in (6). When $\zeta = 1$, the above reduces to SAM (Corollary A.2 of Appendix A.1).

## 4 EXPERIMENTS

In this section, we evaluate the proposed AE-SAM and AE-LookSAM on several standard benchmarks. As the SAM update doubles the computational overhead compared to the ERM update, the training speed is mainly determined by how often the SAM update is used. Hence, we evaluate efficiency by measuring the fraction of SAM updates used: $\%SAM \equiv 100 \times \#\{\text{iterations using SAM}\}/T$. The total number of iterations, $T$, is the same for all methods.

### 4.1 *CIFAR-10* AND *CIFAR-100*

**Setup.** In this section, experiments are performed on the *CIFAR-10* and *CIFAR-100* datasets (Krizhevsky & Hinton, 2009) using four network architectures: *ResNet-18* (He et al., 2016), *WideResNet-28-10* (denoted *WRN-28-10*) (Zagoruyko & Komodakis, 2016), *PyramidNet-110* (Han et al., 2017), and *ViT-S16* (Dosovitskiy et al., 2021).

Following the setup in (Liu et al., 2022; Foret et al., 2021; Zhao et al., 2022a), we use batch size $128$, initial learning rate of $0.1$, cosine learning rate schedule, SGD optimizer with momentum $0.9$ and weight decay $0.0001$. The number of training epochs is $300$ for *PyramidNet-110*, $1200$ for *ViT-S16*, and $200$ for *ResNet-18* and *WideResNet-28-10*. $10\%$ of the training set is used as the validation set. As in Foret et al. (2021), we perform grid search for the radius $\rho$ over $\{0.01, 0.02, 0.05, 0.1, 0.2, 0.5\}$ using the validation set. Similarly, $\alpha$ is selected by grid search over $\{0.1, 0.3, 0.6, 0.9\}$. For the $c_t$ schedule $g_{\lambda_1, \lambda_2}(t)$, $\lambda_1 = -1$ and $\lambda_2 = 1$ for AE-SAM; $\lambda_1 = 0$ and $\lambda_2 = 2$ for AE-LookSAM.

**Baselines.** The proposed AE-SAM and AE-LookSAM are compared with the following baselines: (i) ERM;   (ii) SAM (Foret et al., 2021); and its more efficient variants including   (iii) ESAM (Du et al., 2022a) which uses part of the weights to compute the perturbation and part of the samples to compute the SAM update direction. These two techniques can reduce the computational cost, but may not always accelerate SAM, particularly in parallel training (Li et al., 2020);   (iv) SS-SAM (Zhao et al., 2022b), which randomly selects SAM or ERM according to a Bernoulli trial with success probability $0.5$. This is the scheme with the best performance in (Zhao et al., 2022b);   (v) Look-SAM (Liu et al., 2022) which uses SAM at every $k = 5$ steps. The experiment is repeated five times with different random seeds.

**Results.** Table 1 shows the testing accuracy and fraction of SAM updates (%SAM). Methods are grouped based on %SAM. As can be seen, AE-SAM has higher accuracy than SAM while using only $50\%$ of SAM updates. SS-SAM and AE-SAM have comparable %SAM (about $50\%$), and AE-SAM achieves higher accuracy than SS-SAM (which is statistically significant based on the pairwise t-test at $95\%$ significance level). Finally, LookSAM and AE-LookSAM have comparable %SAM (about $20\%$), and AE-LookSAM also has higher accuracy than LookSAM. These improvements confirm that the adaptive policy is better.

### 4.2 *ImageNet*

**Setup.** In this section, we perform experiments on the *ImageNet* (Russakovsky et al., 2015), which contains $1000$ classes and $1.28$ million images. The *ResNet-50* (He et al., 2016) is used. Following the setup in Du et al. (2022a), we train the network for $90$ epochs using a SGD optimizer with momentum $0.9$, weight decay $0.0001$, initial learning rate $0.1$, cosine learning rate schedule, and batch size $512$. As in (Foret et al., 2021; Du et al., 2022a), $\rho = 0.05$. For the $c_t$ schedule $g_{\lambda_1, \lambda_2}(t)$, $\lambda_1 = -1$ and $\lambda_2 = 1$ for AE-SAM; $\lambda_1 = 0$ and $\lambda_2 = 2$ for AE-LookSAM. $k = 5$ is used for LookSAM. Experiments are repeated with three different random seeds.

**Results.** Table 2 shows the testing accuracy and fraction of SAM updates. As can be seen, with only half of the iterations using SAM, AE-SAM achieves comparable performance as SAM. Compared with LookSAM, AE-LookSAM has better performance (which is also statistically significant), verifying the proposed adaptive policy is more effective than LookSAM's periodic policy.

### 4.3 ROBUSTNESS TO LABEL NOISE

**Setup.** In this section, we study whether the more-efficient SAM variants will affect its robustness to training label noise. Following the setup in Foret et al. (2021), we conduct experiments on a corrupted

Table 1: Means and standard deviations of testing accuracy and fraction of SAM updates (%SAM) on *CIFAR-10* and *CIFAR-100*. Methods are grouped based on %SAM. The highest accuracy in each group is underlined; while the highest accuracy for each network architecture (across all groups) is in bold.

| | | CIFAR-10 | | CIFAR-100 | |
| --- | --- | --- | --- | --- | --- |
| | | Accuracy | %SAM | Accuracy | %SAM |
| **ResNet-18** | ERM | $95.41_{\pm0.03}$ | $0.0_{\pm0.0}$ | $78.17_{\pm0.05}$ | $0.0_{\pm0.0}$ |
| | SAM (Foret et al., 2021) | $96.52_{\pm0.12}$ | $100.0_{\pm0.0}$ | $80.17_{\pm0.15}$ | $100.0_{\pm0.0}$ |
| | ESAM (Du et al., 2022a) | $96.56_{\pm0.08}$ | $100.0_{\pm0.0}$ | $80.41_{\pm0.10}$ | $100.0_{\pm0.0}$ |
| | SS-SAM (Zhao et al., 2022b) | $96.40_{\pm0.16}$ | $50.0_{\pm0.0}$ | $80.10_{\pm0.16}$ | $50.0_{\pm0.0}$ |
| | AE-SAM | $\underline{\mathbf{96.63}}_{\pm0.04}$ | $50.1_{\pm0.1}$ | $\underline{\mathbf{80.48}}_{\pm0.11}$ | $49.8_{\pm0.0}$ |
| | LookSAM (Liu et al., 2022) | $96.32_{\pm0.12}$ | $20.0_{\pm0.0}$ | $79.89_{\pm0.29}$ | $20.0_{\pm0.0}$ |
| | AE-LookSAM | $\underline{96.56}_{\pm0.21}$ | $20.0_{\pm0.1}$ | $\underline{80.29}_{\pm0.37}$ | $20.0_{\pm0.0}$ |
| **WRN-28-10** | ERM | $96.34_{\pm0.12}$ | $0.0_{\pm0.0}$ | $81.56_{\pm0.14}$ | $0.0_{\pm0.0}$ |
| | SAM (Foret et al., 2021) | $97.27_{\pm0.11}$ | $100.0_{\pm0.0}$ | $83.42_{\pm0.05}$ | $100.0_{\pm0.0}$ |
| | ESAM (Du et al., 2022a) | $97.29_{\pm0.11}$ | $100.0_{\pm0.0}$ | $\mathbf{84.51}_{\pm0.02}$ | $100.0_{\pm0.0}$ |
| | SS-SAM (Zhao et al., 2022b) | $97.09_{\pm0.11}$ | $50.0_{\pm0.0}$ | $82.89_{\pm0.02}$ | $50.0_{\pm0.0}$ |
| | AE-SAM | $\underline{\mathbf{97.30}}_{\pm0.10}$ | $49.5_{\pm0.1}$ | $\underline{\mathbf{84.51}}_{\pm0.11}$ | $49.6_{\pm0.0}$ |
| | LookSAM (Liu et al., 2022) | $97.02_{\pm0.12}$ | $20.0_{\pm0.0}$ | $83.70_{\pm0.12}$ | $20.0_{\pm0.0}$ |
| | AE-LookSAM | $\underline{97.15}_{\pm0.08}$ | $20.0_{\pm0.0}$ | $\underline{83.92}_{\pm0.07}$ | $20.2_{\pm0.0}$ |
| **PyramidNet-110** | ERM | $96.62_{\pm0.10}$ | $0.0_{\pm0.0}$ | $81.89_{\pm0.15}$ | $0.0_{\pm0.0}$ |
| | SAM (Foret et al., 2021) | $97.30_{\pm0.10}$ | $100.0_{\pm0.0}$ | $84.46_{\pm0.05}$ | $100.0_{\pm0.0}$ |
| | ESAM (Du et al., 2022a) | $97.81_{\pm0.01}$ | $100.0_{\pm0.0}$ | $85.56_{\pm0.05}$ | $100.0_{\pm0.0}$ |
| | SS-SAM (Zhao et al., 2022b) | $97.22_{\pm0.10}$ | $50.0_{\pm0.0}$ | $84.90_{\pm0.05}$ | $50.0_{\pm0.0}$ |
| | AE-SAM | $\underline{\mathbf{97.90}}_{\pm0.05}$ | $50.2_{\pm0.1}$ | $\underline{\mathbf{85.58}}_{\pm0.10}$ | $49.8_{\pm0.1}$ |
| | LookSAM (Liu et al., 2022) | $97.10_{\pm0.11}$ | $20.0_{\pm0.0}$ | $84.01_{\pm0.06}$ | $20.0_{\pm0.0}$ |
| | AE-LookSAM | $\underline{97.22}_{\pm0.11}$ | $20.3_{\pm0.0}$ | $\underline{84.80}_{\pm0.13}$ | $20.2_{\pm0.1}$ |
| **ViT-S16** | ERM | $86.69_{\pm0.11}$ | $0.0_{\pm0.0}$ | $62.42_{\pm0.22}$ | $0.0_{\pm0.0}$ |
| | SAM (Foret et al., 2021) | $87.37_{\pm0.09}$ | $100.0_{\pm0.0}$ | $63.23_{\pm0.25}$ | $100.0_{\pm0.0}$ |
| | ESAM (Du et al., 2022a) | $84.27_{\pm0.11}$ | $100.0_{\pm0.0}$ | $62.11_{\pm0.15}$ | $100.0_{\pm0.0}$ |
| | SS-SAM (Zhao et al., 2022b) | $87.38_{\pm0.14}$ | $50.0_{\pm0.0}$ | $63.18_{\pm0.19}$ | $50.0_{\pm0.0}$ |
| | AE-SAM | $\underline{\mathbf{87.77}}_{\pm0.13}$ | $49.7_{\pm0.1}$ | $\underline{63.68}_{\pm0.23}$ | $49.5_{\pm0.2}$ |
| | LookSAM (Liu et al., 2022) | $87.12_{\pm0.20}$ | $20.0_{\pm0.0}$ | $63.52_{\pm0.19}$ | $20.0_{\pm0.0}$ |
| | AE-LookSAM | $\underline{87.32}_{\pm0.11}$ | $20.2_{\pm0.2}$ | $\underline{\mathbf{64.16}}_{\pm0.23}$ | $20.3_{\pm0.2}$ |

version of *CIFAR-10*, with some of its training labels randomly flipped (while its testing set is kept clean). The *ResNet-18* and *ResNet-32* networks are used. They are trained for 200 epochs using SGD with momentum 0.9, weight decay 0.0001, batch size 128, initial learning rate 0.1, and cosine learning rate schedule. For LookSAM, the SAM update is used every $k = 2$ steps.[3] For AE-SAM and AE-LookSAM, we set $\lambda_1 = -1$ and $\lambda_2 = 1$ in their $c_t$ schedules $g_{\lambda_1, \lambda_2}(t)$, such that their fractions of SAM updates (approximately $50\%$) are comparable with SS-SAM and LookSAM. Experiments are repeated with five different random seeds.

**Results.** Table 3 shows the testing accuracy and fraction of SAM updates. As can be seen, AE-LookSAM achieves comparable performance with SAM but is faster as only half of the iterations use the SAM update. Compared with ESAM, SS-SAM, and LookSAM, AE-LookSAM performs better. The improvement is particularly noticeable at the higher noise levels (e.g., $80\%$).

---

[3]The performance of LookSAM can be sensitive to the value of $k$. Table 4 of Appendix B.2 shows that using $k = 2$ leads to the best performance in this experiment.

Table 2: Means and standard deviations of testing accuracy and fraction of SAM updates (%SAM) on *ImageNet* using *ResNet-50*. Methods are grouped based on %SAM. The highest accuracy in each group is underlined; while the highest across all groups is in bold.

| | Accuracy | %SAM |
|---|---|---|
| ERM | 77.11 ±0.14 | 0.0 ±0.0 |
| SAM (Foret et al., 2021) | **77.47** ±0.12 | 100.0 ±0.0 |
| ESAM (Du et al., 2022a) | 77.25 ±0.75 | 100.0 ±0.0 |
| SS-SAM (Zhao et al., 2022b) | 77.38 ±0.06 | 50.0 ±0.0 |
| AE-SAM | 77.43 ±0.06 | 49.4 ±0.0 |
| LookSAM (Liu et al., 2022) | 77.13 ±0.09 | 20.0 ±0.0 |
| AE-LookSAM | 77.29 ±0.08 | 20.3 ±0.0 |

Table 3: Testing accuracy and fraction of SAM updates on *CIFAR-10* with different levels of label noise. The best accuracy is in bold and the second best is underlined.

| | | noise = 20% | | noise = 40% | | noise = 60% | | noise = 80% | |
|---|---|---|---|---|---|---|---|---|---|
| | | accuracy | %SAM | accuracy | %SAM | accuracy | %SAM | accuracy | %SAM |
| *ResNet-18* | ERM | 87.92 | 0.0 | 70.82 | 0.0 | 49.61 | 0.0 | 28.23 | 0.0 |
| | SAM (Foret et al., 2021) | **94.80** | 100.0 | 91.50 | 100.0 | **88.15** | 100.0 | **77.40** | 100.0 |
| | ESAM (Du et al., 2022a) | 94.19 | 100.0 | 91.46 | 100.0 | 81.30 | 100.0 | 15.00 | 100.0 |
| | SS-SAM (Zhao et al., 2022b) | 90.62 | 50.0 | 77.84 | 50.0 | 61.18 | 50.0 | 47.32 | 50.0 |
| | LookSAM (Liu et al., 2022) | 92.72 | 50.0 | 88.04 | 50.0 | 72.26 | 50.0 | 69.72 | 50.0 |
| | AE-SAM | 92.84 | 50.0 | 84.17 | 50.0 | 73.54 | 49.9 | 65.00 | 50.0 |
| | AE-LookSAM | 94.34 | 49.9 | **91.58** | 50.0 | 87.85 | 50.0 | 76.90 | 50.0 |
| *ResNet-32* | ERM | 87.43 | 0.0 | 70.82 | 0.0 | 46.26 | 0.0 | 29.00 | 0.0 |
| | SAM (Foret et al., 2021) | **95.08** | 100.0 | 91.01 | 100.0 | **88.90** | 100.0 | **77.32** | 100.0 |
| | ESAM (Du et al., 2022a) | 93.42 | 100.0 | 91.63 | 100.0 | 82.73 | 100.0 | 10.09 | 100.0 |
| | SS-SAM (Zhao et al., 2022b) | 89.63 | 50.0 | 74.17 | 50.0 | 58.40 | 50.0 | 59.53 | 50.0 |
| | LookSAM (Liu et al., 2022) | 92.49 | 50.0 | 86.56 | 50.0 | 63.35 | 50.0 | 68.01 | 50.0 |
| | AE-SAM | 92.87 | 50.0 | 82.85 | 50.0 | 71.50 | 50.0 | 65.43 | 50.3 |
| | AE-LookSAM | 94.70 | 50.0 | **91.80** | 50.0 | 88.22 | 50.0 | 77.03 | 49.8 |

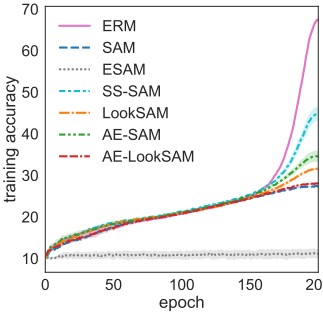

(a) Training accuracy.

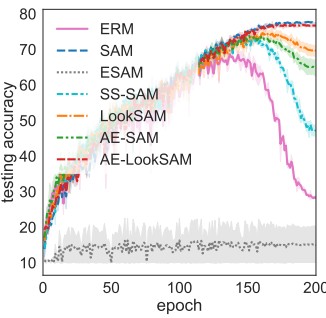

(b) Testing accuracy.

Figure 4: Accuracies with number of training epochs on *CIFAR-10* (with 80% noise labels) using *ResNet-18*. Best viewed in color.

Figure 4 shows the training and testing accuracies with number of epochs at a noise level of 80% using *ResNet-18*[4]. As can be seen, SAM is robust to the label noise, while ERM and SS-SAM heavily suffer from overfitting. AE-SAM and LookSAM can alleviate the overfitting problem to a certain extent. AE-LookSAM, by combining the adaptive policy with LookSAM, achieves the same high level of robustness as SAM.

---

[4]Results for other noise levels and *ResNet-32* are shown in Figures 10 and 11 of Appendix B.3, respectively.

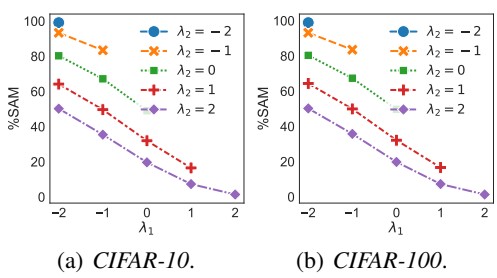

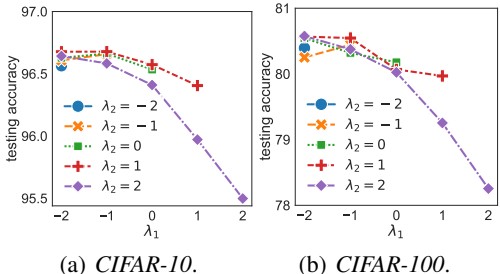

Figure 5: Effects of $\lambda_1$ and $\lambda_2$ on fraction of SAM updates using *ResNet-18*. Best viewed in color.

Figure 6: Effects of $\lambda_1$ and $\lambda_2$ on testing accuracy using *ResNet-18*. Best viewed in color.

### 4.4 EFFECTS OF $\lambda_1$ AND $\lambda_2$

In this experiment, we study the effects of $\lambda_1$ and $\lambda_2$ on AE-SAM. We use the same setup as in Section 4.1, where $\lambda_1$ and $\lambda_2$ (with $\lambda_1 \leq \lambda_2$) are chosen from $\{0, \pm 1, \pm 2\}$. Results on AE-LookSAM using the label noise setup in Section 4.3 are shown in Appendix B.4.

Figure 5 shows the effect on the fraction of SAM updates. For a fixed $\lambda_2$, increasing $\lambda_1$ increases the threshold $c_t$, and the condition $\|\nabla \mathcal{L}(\mathcal{B}_t; \mathbf{w}_t)\|^2 \geq \mu_t + c_t \sigma_t$ becomes more difficult to satisfy. Thus, as can be seen, the fraction of SAM updates is reduced. The same applies when $\lambda_2$ increases. A similar trend is also observed on the testing accuracy (Figure 6).

### 4.5 CONVERGENCE

In this experiment, we study whether $\mathbf{w}_t$'s (where $t$ is the number of epochs) obtained from AE-SAM can reach critical points of $\mathcal{L}(\mathcal{D}; \mathbf{w})$, as suggested in Theorem 3.3. Figure 7 shows $\|\nabla \mathcal{L}(\mathcal{D}; \mathbf{w}_t)\|^2$ w.r.t. $t$ for the experiment in Section 4.1. As can be seen, in all settings, $\|\nabla \mathcal{L}(\mathcal{D}; \mathbf{w}_t)\|^2$ converges to 0. In Appendix B.5, we also verify the convergence of AE-SAM's training loss on *CIFAR-10* and *CIFAR-100* (Figure 14), and that AE-SAM and SS-SAM have comparable convergence speeds (Figure 15), which agrees with Theorem 3.3 as both have comparable fractions of SAM updates (Table 1).

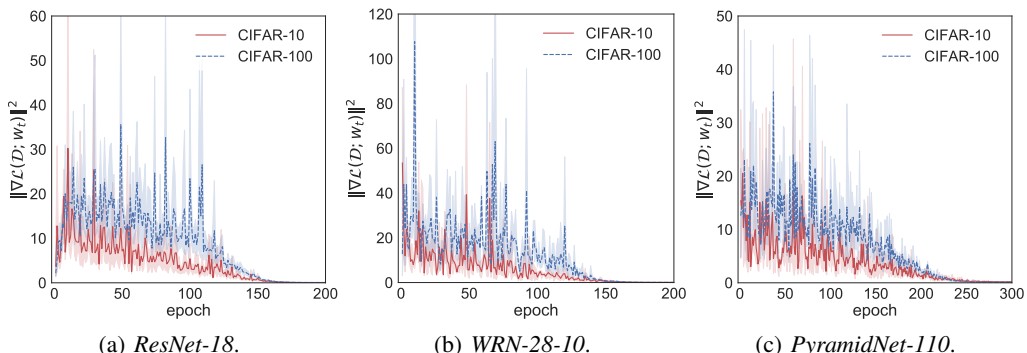

Figure 7: Squared gradient norms of AE-SAM with number of epochs. Best viewed in color.

## 5 CONCLUSION

In this paper, we proposed an adaptive policy to employ SAM based on the loss landscape geometry. Using the policy, we proposed an efficient algorithm (called AE-SAM) to reduce the fraction of SAM updates during training. We theoretically and empirically analyzed the convergence of AE-SAM. Experimental results on a number of datasets and network architectures verify the efficiency and effectiveness of the adaptive policy. Moreover, the proposed policy is general and can be combined with other SAM variants, as demonstrated by the success of AE-LookSAM.

ACKNOWLEDGMENTS

This work was supported by NSFC key grant 62136005, NSFC general grant 62076118, and Shenzhen fundamental research program JCYJ20210324105000003. This research was supported in part by the Research Grants Council of the Hong Kong Special Administrative Region (Grant 16200021).

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

## A  PROOFS

### A.1  PROOF OF THEOREM 3.3

**Theorem 3.3.** *Let $b$ be the mini-batch size. If $\eta = \frac{1}{4\beta\sqrt{T}}$ and $\rho = 1/T^{\frac{1}{4}}$, algorithm $\mathcal{A}$ satisfies*

$$\min_{0 \leq t \leq T-1} \mathbb{E}\|\nabla\mathcal{L}(\mathcal{D}; \mathbf{w}_t)\|^2 \leq \frac{32\beta\left(\mathcal{L}(\mathcal{D}; \mathbf{w}_0) - \mathbb{E}\mathcal{L}(\mathcal{D}; \mathbf{w}_T)\right)}{\sqrt{T}\,(7 - 6\zeta)} + \frac{(1 + \zeta + 5\beta^2\zeta)\sigma^2}{b\sqrt{T}\,(7 - 6\zeta)}, \quad (7)$$

*where $\zeta = \frac{1}{T}\sum_{t=0}^{T-1}\xi_t \in [0, 1]$.*

**Lemma A.1** (Andriushchenko & Flammarion (2022))**.** *Under Assumptions 3.1 and 3.2 for all $t$ and $\rho > 0$, we have*

$$\mathbb{E}\nabla\mathcal{L}(\mathcal{B}_t; \mathbf{w} + \rho\nabla\mathcal{L}(\mathcal{B}_t; \mathbf{w}))^\top \nabla\mathcal{L}(\mathcal{D}; \mathbf{w}) \geq \left(\frac{1}{2} - \rho\beta\right)\|\nabla\mathcal{L}(\mathcal{D}; \mathbf{w})\|^2 - \frac{\beta^2\rho^2\sigma^2}{2b}. \quad (8)$$

*Proof.* Let $\mathbf{g}_t \equiv \frac{1}{b}\sum_{(\mathbf{x}_i, y_i) \in \mathcal{B}_t} \nabla\ell(f(\mathbf{x}_i; \mathbf{w}_t), y_i)$, $\mathbf{h}_t \equiv \frac{1}{b}\sum_{(\mathbf{x}_i, y_i) \in \mathcal{B}_t} \nabla\ell(f(\mathbf{x}_i; \mathbf{w}_t + \rho\mathbf{g}_t), y_i)$, and $\hat{\mathbf{g}}_t \equiv \nabla\mathcal{L}(\mathcal{D}; \mathbf{w}_t)$.

By Taylor expansion and $\mathcal{L}(\mathcal{D}; \mathbf{w})$ is $\beta$-smooth, we have

$$\mathcal{L}(\mathcal{D}; \mathbf{w}_{t+1})$$

$$\leq \mathcal{L}(\mathcal{D}; \mathbf{w}_t) + \hat{\mathbf{g}}_t^\top(\mathbf{w}_{t+1} - \mathbf{w}_t) + \frac{\beta}{2}\|\mathbf{w}_{t+1} - \mathbf{w}_t\|^2$$

$$\leq \mathcal{L}(\mathcal{D}; \mathbf{w}_t) - \eta\hat{\mathbf{g}}_t^\top\left((1 - \xi_t)\mathbf{g}_t + \xi_t\mathbf{h}_t\right) + \frac{\beta\eta^2}{2}\|(1 - \xi_t)\mathbf{g}_t + \xi_t\mathbf{h}_t\|^2$$

$$= \mathcal{L}(\mathcal{D}; \mathbf{w}_t) - \eta(1 - \xi_t)\hat{\mathbf{g}}_t^\top\mathbf{g}_t - \eta\xi_t\hat{\mathbf{g}}_t^\top\mathbf{h}_t + \frac{\beta\eta^2}{2}\left((1 - \xi_t)\|\mathbf{g}_t\|^2 + \xi_t\|\mathbf{h}_t\|^2 + \underbrace{2\xi_t(1 - \xi_t)\mathbf{g}_t^\top\mathbf{h}_t}_{=0}\right) \quad (9)$$

$$= \mathcal{L}(\mathcal{D}; \mathbf{w}_t) - \eta(1 - \xi_t)\hat{\mathbf{g}}_t^\top\mathbf{g}_t - \eta\xi_t\hat{\mathbf{g}}_t^\top\mathbf{h}_t + \frac{\beta\eta^2}{2}\left((1 - \xi_t)\|\mathbf{g}_t\|^2 + \xi_t\|\mathbf{h}_t\|^2\right), \quad (10)$$

where we have used $\xi_t(1 - \xi_t) = 0$ as $\xi_t \in \{0, 1\}$, $\xi_t^2 = \xi_t$, and $(1 - \xi_t)^2 = 1 - \xi_t$ to obtain (9). Taking expectation w.r.t. $\mathbf{w}_t$ on both sides of (10), we have

$$\mathbb{E}\mathcal{L}(\mathcal{D}; \mathbf{w}_{t+1}) \leq \mathbb{E}\mathcal{L}(\mathcal{D}; \mathbf{w}_t) - \eta(1-\xi_t)\mathbb{E}\|\hat{\mathbf{g}}_t\|^2 - \eta\xi_t\mathbb{E}\hat{\mathbf{g}}_t^\top \mathbf{h}_t + \frac{\beta\eta^2(1-\xi_t)}{2}\mathbb{E}\|\mathbf{g}_t\|^2 + \frac{\beta\eta^2\xi_t}{2}\mathbb{E}\|\mathbf{h}_t\|^2. \tag{11}$$

Claim 1: $\mathbb{E}\|\mathbf{g}_t\|^2 = \mathbb{E}\|\mathbf{g}_t - \hat{\mathbf{g}}_t\|^2 + \mathbb{E}\|\hat{\mathbf{g}}_t\|^2 = \frac{\sigma^2}{b} + \mathbb{E}\|\hat{\mathbf{g}}_t\|^2$, which follows from Assumption 3.2.

Claim 2: $\mathbb{E}\|\mathbf{h}_t\|^2 \leq 2(1 + \rho^2\beta^2)\frac{\sigma^2}{b} - (1 - 2\rho^2\beta^2)\mathbb{E}\|\hat{\mathbf{g}}_t\|^2 + 2\mathbb{E}\hat{\mathbf{g}}_t^\top \mathbf{h}_t$, which is derived as follows:

$$\begin{aligned}
\mathbb{E}\|\mathbf{h}_t\|^2 &= \mathbb{E}\|\mathbf{h}_t - \hat{\mathbf{g}}_t\|^2 - \mathbb{E}\|\hat{\mathbf{g}}_t\|^2 + 2\mathbb{E}\hat{\mathbf{g}}_t^\top \mathbf{h}_t \\
&= 2\mathbb{E}\|\mathbf{h}_t - \mathbf{g}_t\|^2 + 2\mathbb{E}\|\mathbf{g}_t - \hat{\mathbf{g}}_t\|^2 - \mathbb{E}\|\hat{\mathbf{g}}_t\|^2 + 2\mathbb{E}\hat{\mathbf{g}}_t^\top \mathbf{h}_t \\
&\leq 2\rho^2\beta^2\mathbb{E}\|\mathbf{g}_t\|^2 + \frac{2\sigma^2}{b} - \mathbb{E}\|\hat{\mathbf{g}}_t\|^2 + 2\mathbb{E}\hat{\mathbf{g}}_t^\top \mathbf{h}_t \tag{12} \\
&\leq 2\rho^2\beta^2 \left(\frac{\sigma^2}{b} + \mathbb{E}\|\hat{\mathbf{g}}_t\|^2\right) + \frac{2\sigma^2}{b} - \mathbb{E}\|\hat{\mathbf{g}}_t\|^2 + 2\mathbb{E}\hat{\mathbf{g}}_t^\top \mathbf{h}_t \tag{13} \\
&= 2(1 + \rho^2\beta^2)\frac{\sigma^2}{b} - (1 - 2\rho^2\beta^2)\mathbb{E}\|\hat{\mathbf{g}}_t\|^2 + 2\mathbb{E}\hat{\mathbf{g}}_t^\top \mathbf{h}_t, \tag{14}
\end{aligned}$$

where (12) follows from $\|\mathbf{h}_t - \mathbf{g}_t\| \leq \rho\beta\|\mathbf{g}_t\|$ and Assumption 3.2, (13) follows from Claim 1. Substituting Claims 1 and 2 into (11), we obtain

$$\begin{aligned}
&\mathbb{E}\mathcal{L}(\mathcal{D}; \mathbf{w}_{t+1}) \\
&\leq \mathbb{E}\mathcal{L}(\mathcal{D}; \mathbf{w}_t) - \eta\left(1 - \xi_t\right)\mathbb{E}\|\hat{\mathbf{g}}_t\|^2 - \eta\xi_t\mathbb{E}\hat{\mathbf{g}}_t^\top \mathbf{h}_t + \frac{\beta\eta^2(1-\xi_t)}{2}\left(\frac{\sigma^2}{b} + \mathbb{E}\|\hat{\mathbf{g}}_t\|^2\right) \\
&\quad + \frac{\beta\eta^2\xi_t}{2}\left(2(1 + \rho^2\beta^2)\frac{\sigma^2}{b} - (1 - 2\rho^2\beta^2)\mathbb{E}\|\hat{\mathbf{g}}_t\|^2 + 2\mathbb{E}\hat{\mathbf{g}}_t^\top \mathbf{h}_t\right) \tag{15} \\
&= \mathbb{E}\mathcal{L}(\mathcal{D}; \mathbf{w}_t) - \eta\left(1 - \xi_t - \frac{\beta\eta(1-\xi_t)}{2} + \frac{\beta\eta\xi_t(1 - 2\rho^2\beta^2)}{2}\right)\mathbb{E}\|\hat{\mathbf{g}}_t\|^2 - \eta\xi_t\left(1 - \eta\beta\right)\mathbb{E}\hat{\mathbf{g}}_t^\top \mathbf{h}_t \\
&\quad + \left(\frac{\beta\eta^2(1-\xi_t)}{2} + \beta\eta^2\xi_t(1 + \rho^2\beta^2)\right)\frac{\sigma^2}{b} \\
&\leq \mathbb{E}\mathcal{L}(\mathcal{D}; \mathbf{w}_t) - \eta\left(1 - \xi_t - \frac{\beta\eta(1-\xi_t)}{2} + \frac{\beta\eta\xi_t(1 - 2\rho^2\beta^2)}{2} + \xi_t\left(1 - \eta\beta\right)\left(\frac{1}{2} - \rho\beta\right)\right)\mathbb{E}\|\hat{\mathbf{g}}_t\|^2 \\
&\quad + \left(\frac{\beta\eta^2(1-\xi_t)}{2} + \beta\eta^2\xi_t(1 + \rho^2\beta^2) + \eta\xi_t\left(1 - \eta\beta\right)\frac{\beta^2\rho^2}{2}\right)\frac{\sigma^2}{b} \tag{16} \\
&\leq \mathbb{E}\mathcal{L}(\mathcal{D}; \mathbf{w}_t) - \eta\left(1 - (1 + \beta\eta - 2\rho\beta)\frac{\xi_t}{2} - \frac{\beta\eta}{2}\right)\mathbb{E}\|\hat{\mathbf{g}}_t\|^2 \\
&\quad + \left(\eta + \xi_t(\eta + 2\eta\rho^2\beta^2 + \beta\rho^2 - \eta\beta^2\rho^2)\right)\frac{\eta\beta\sigma^2}{2b}, \tag{17}
\end{aligned}$$

where (15) follows from Claims 1 and 2, (16) follows from Lemma A.1 and $1 - \eta\beta > 0$. As $\eta < \frac{1}{4\beta}$, we have $1 + \beta\eta - 2\rho\beta \leq 3/2$ and $\beta\eta < 1/4$, thus, $1 - (1 + \beta\eta - 2\rho\beta)\frac{\xi_t}{2} - \frac{\beta\eta}{2} > 0$.

Summing over $t$ on both sides of (17) and rearranging, we obtain

$$
\min_{0 \le t \le T-1} \mathbb{E}\|\hat{\mathbf{g}}_t\|^2 \le \frac{\mathcal{L}(\mathcal{D}; \mathbf{w}_0) - \mathbb{E}\mathcal{L}(\mathcal{D}; \mathbf{w}_T)}{\eta \sum_{t=0}^{T-1}\left(1 - (1+\beta\eta - 2\rho\beta)\frac{\xi_t}{2} - \frac{\beta\eta}{2}\right)}
$$

$$
+ \frac{\sum_{t=0}^{T-1}\left(\eta + \xi_t(\eta + \eta\rho^2\beta^2 + \beta\rho^2)\right)}{\sum_{t=0}^{T-1}\left(1 - (1+\beta\eta - 2\rho\beta)\frac{\xi_t}{2} - \frac{\beta\eta}{2}\right)} \frac{\beta\sigma^2}{2b}
$$

$$
= \frac{\mathcal{L}(\mathcal{D}; \mathbf{w}_0) - \mathbb{E}\mathcal{L}(\mathcal{D}; \mathbf{w}_T)}{T\eta(1 - \frac{\gamma\zeta}{2} - \frac{\beta\eta}{2})} + \frac{T(\eta + \eta\kappa\zeta + \beta\rho^2\zeta)\beta\sigma^2}{2bT(1 - \frac{\gamma\zeta}{2} - \frac{\beta\eta}{2})} \qquad (18)
$$

$$
= \frac{\mathcal{L}(\mathcal{D}; \mathbf{w}_0) - \mathbb{E}\mathcal{L}(\mathcal{D}; \mathbf{w}_T)}{T\eta(1 - \frac{\gamma\zeta}{2} - \frac{\beta\eta}{2})} + \frac{(1 + \kappa\zeta + 4\beta^2\zeta)\eta\beta\sigma^2}{2b(1 - \frac{\gamma\zeta}{2} - \frac{\beta\eta}{2})}
$$

$$
= \frac{\mathcal{L}(\mathcal{D}; \mathbf{w}_0) - \mathbb{E}\mathcal{L}(\mathcal{D}; \mathbf{w}_T)}{T\eta(1 - \frac{\gamma\zeta}{2} - \frac{\beta\eta}{2})} + \frac{(1 + \kappa\zeta + 4\beta^2\zeta)\sigma^2}{8b\sqrt{T}(1 - \frac{\gamma\zeta}{2} - \frac{\beta\eta}{2})} \qquad (19)
$$

$$
\le \frac{32\beta\left(\mathcal{L}(\mathcal{D}; \mathbf{w}_0) - \mathbb{E}\mathcal{L}(\mathcal{D}; \mathbf{w}_T)\right)}{\sqrt{T}\,(7 - 6\zeta)} + \frac{(1 + \zeta + 5\beta^2\zeta)\sigma^2}{b\sqrt{T}\,(7 - 6\zeta)}, \qquad (20)
$$

where $\gamma = 1 + \beta\eta - 2\rho\beta \le 3/2$, $\kappa = 1 + \rho^2\beta^2$, $\rho^2 = 1/\sqrt{T}$, and $\zeta = \frac{1}{T}\sum_{t=0}^{T-1}\xi_t \in [0,1]$. We thus finish the proof. $\qquad\square$

**Corollary A.2.** *Let $b$ be the mini-batch size. If $\eta = \frac{1}{4\beta\sqrt{T}}$ and $\rho = 1/T^{\frac{1}{4}}$, SAM (Foret et al., 2021) satisfies*

$$
\min_{0 \le t \le T-1} \mathbb{E}\|\nabla\mathcal{L}(\mathcal{D}; \mathbf{w}_t)\|^2 \le \frac{32\beta\left(\mathcal{L}(\mathcal{D}; \mathbf{w}_0) - \mathbb{E}\mathcal{L}(\mathcal{D}; \mathbf{w}_T)\right)}{\sqrt{T}} + \frac{(2 + 5\beta^2)\sigma^2}{b\sqrt{T}}. \qquad (21)
$$

**Corollary A.3.** *Let $b$ be the mini-batch size. If $\eta = \frac{\sqrt{b}}{4\beta\sqrt{T}}$ and $\rho = 1/T^{\frac{1}{4}}$, algorithm $\mathcal{A}$ satisfies*

$$
\min_{0 \le t \le T-1} \mathbb{E}\|\nabla\mathcal{L}(\mathcal{D}; \mathbf{w}_t)\|^2 \le \frac{32\beta(\mathcal{L}(\mathcal{D}; \mathbf{w}_0) - \mathbb{E}\mathcal{L}(\mathcal{D}; \mathbf{w}_T))}{\sqrt{Tb}(7 - 6\zeta)} + \frac{(1 + \zeta + 5\beta^2\zeta)\sigma^2}{\sqrt{Tb}(7 - 6\zeta)}, \qquad (22)
$$

*where $\zeta = \frac{1}{T}\sum_{t=0}^{T-1}\xi_t \in [0,1]$.*

*Proof.* It follows from (18) that

$$
\min_{0 \le t \le T-1} \mathbb{E}\|\hat{\mathbf{g}}_t\|^2 \le \frac{\mathcal{L}(\mathcal{D}; \mathbf{w}_0) - \mathbb{E}\mathcal{L}(\mathcal{D}; \mathbf{w}_T)}{T\eta(1 - \frac{\gamma\zeta}{2} - \frac{\beta\eta}{2})} + \frac{\eta\beta(1 + \kappa\zeta + 4\beta^2\zeta)\sigma^2}{2b(1 - \frac{\gamma\zeta}{2} - \frac{\beta\eta}{2})} \qquad (23)
$$

$$
\le \frac{4\beta(\mathcal{L}(\mathcal{D}; \mathbf{w}_0) - \mathbb{E}\mathcal{L}(\mathcal{D}; \mathbf{w}_T))}{\sqrt{Tb}(\frac{7}{8} - \frac{3}{4}\zeta)} + \frac{(1 + \zeta + 5\beta^2\zeta)\sigma^2}{8\sqrt{Tb}(\frac{7}{8} - \frac{3\zeta}{4})} \qquad (24)
$$

$$
= \frac{32\beta(\mathcal{L}(\mathcal{D}; \mathbf{w}_0) - \mathbb{E}\mathcal{L}(\mathcal{D}; \mathbf{w}_T))}{\sqrt{Tb}(7 - 6\zeta)} + \frac{(1 + \zeta + 5\beta^2\zeta)\sigma^2}{\sqrt{Tb}(7 - 6\zeta)}. \qquad (25)
$$

$$\square$$

## A.2 CONVERGENCE OF FULL-BATCH GRADIENT DESCENT FOR AE-SAM

**Theorem A.4.** *Under Assumption 3.1, with full-batch gradient descent, if $\rho < \frac{1}{2\beta}$ and $\eta < \frac{1}{\beta}$, algorithm $\mathcal{A}$ satisfies*

$$
\min_{0 \le t \le T-1} \|\nabla\mathcal{L}(\mathcal{D}; \mathbf{w}_t)\|^2 \le \frac{\mathcal{L}(\mathcal{D}; \mathbf{w}_0) - \mathcal{L}(\mathcal{D}; \mathbf{w}_T)}{T\eta\left(1 - \frac{\beta\eta}{2} - \beta\rho\zeta\right)}, \qquad (26)
$$

*where $\zeta = \frac{1}{T}\sum_{t=0}^{T-1}\xi_t \in [0,1]$.*

**Lemma A.5** (Lemma 7 in Andriushchenko & Flammarion (2022)). *Let $\mathcal{L}(\mathcal{D}; \mathbf{w})$ be a $\beta$-smooth function. For any $\rho > 0$, we have*

$$\nabla\mathcal{L}(\mathcal{D}; \mathbf{w})^\top \nabla\mathcal{L}(\mathcal{D}; \mathbf{w} + \rho\nabla\mathcal{L}(\mathcal{D}; \mathbf{w})) \geq (1 - \rho\beta)\|\nabla\mathcal{L}(\mathcal{D}; \mathbf{w})\|^2. \tag{27}$$

*Proof of Theorem A.4.* Let $\mathbf{g}_t \equiv \nabla\mathcal{L}(\mathcal{D}; \mathbf{w}_t)$ and $\mathbf{h}_t \equiv \nabla\mathcal{L}(\mathcal{D}; \mathbf{w}_t + \rho\nabla\mathcal{L}(\mathcal{D}; \mathbf{w}_t))$ be the update direction of ERM and SAM, respectively. By Taylor expansion and $\mathcal{L}(\mathcal{D}; \mathbf{w})$ is $\beta$-smooth, we have

$$\mathcal{L}(\mathcal{D}; \mathbf{w}_{t+1})$$

$$\leq \mathcal{L}(\mathcal{D}; \mathbf{w}_t) + \mathbf{g}_t^\top(\mathbf{w}_{t+1} - \mathbf{w}_t) + \frac{\beta}{2}\|\mathbf{w}_{t+1} - \mathbf{w}_t\|^2$$

$$\leq \mathcal{L}(\mathcal{D}; \mathbf{w}_t) - \eta\mathbf{g}_t^\top\left((1-\xi_t)\mathbf{g}_t + \xi_t\mathbf{h}_t\right) + \frac{\beta\eta^2}{2}\|(1-\xi_t)\mathbf{g}_t + \xi_t\mathbf{h}_t\|^2$$

$$= \mathcal{L}(\mathcal{D}; \mathbf{w}_t) - \eta(1-\xi_t)\|\mathbf{g}_t\|^2 - \eta\xi_t\mathbf{g}_t^\top\mathbf{h}_t + \frac{\beta\eta^2}{2}\left((1-\xi_t)\|\mathbf{g}_t\|^2 + \xi_t\|\mathbf{h}_t\|^2 + \underbrace{2\xi_t(1-\xi_t)\mathbf{g}_t^\top\mathbf{h}_t}_{=0}\right) \tag{28}$$

$$= \mathcal{L}(\mathcal{D}; \mathbf{w}_t) - \eta\left(1 - \xi_t - \frac{\beta\eta(1-\xi_t)}{2}\right)\|\mathbf{g}_t\|^2 + \frac{\beta\eta^2\xi_t}{2}\|\mathbf{h}_t\|^2 - \eta\xi_t\mathbf{g}_t^\top\mathbf{h}_t, \tag{29}$$

where we have used $\xi_t(1-\xi_t) = 0$ as $\xi_t \in \{0, 1\}$, $\xi_t^2 = \xi_t$, and $(1-\xi_t)^2 = 1 - \xi_t$ to obtain (28).

As $\|\mathbf{h}_t\|^2 = \|\mathbf{h}_t - \mathbf{g}_t\|^2 - \|\mathbf{g}_t\|^2 + 2\mathbf{g}_t^\top\mathbf{h}_t$, it follows from (29) that

$$\mathcal{L}(\mathcal{D}; \mathbf{w}_{t+1})$$

$$= \mathcal{L}(\mathcal{D}; \mathbf{w}_t) - \eta\left(1 - \xi_t - \frac{\beta\eta(1-\xi_t)}{2}\right)\|\mathbf{g}_t\|^2 + \frac{\beta\eta^2\xi_t}{2}\left(\|\mathbf{h}_t - \mathbf{g}_t\|^2 - \|\mathbf{g}_t\|^2 + 2\mathbf{g}_t^\top\mathbf{h}_t\right) - \eta\xi_t\mathbf{g}_t^\top\mathbf{h}_t$$

$$\leq \mathcal{L}(\mathcal{D}; \mathbf{w}_t) - \eta\left(1 - \xi_t - \frac{\beta\eta(1-\xi_t)}{2} + \frac{\beta\eta\xi_t}{2}\right)\|\mathbf{g}_t\|^2 + \frac{\beta\eta^2\xi_t}{2}\|\mathbf{h}_t - \mathbf{g}_t\|^2 - \eta(1-\beta\eta)\xi_t\mathbf{g}_t^\top\mathbf{h}_t$$

$$\leq \mathcal{L}(\mathcal{D}; \mathbf{w}_t) - \eta\left(1 - \xi_t - \frac{\beta\eta(1-\xi_t)}{2} + \frac{\beta\eta\xi_t}{2}\right)\|\mathbf{g}_t\|^2 + \frac{\beta^3\eta^2\rho^2\xi_t}{2}\|\mathbf{g}_t\|^2 - \eta(1-\beta\eta)\xi_t\mathbf{g}_t^\top\mathbf{h}_t \tag{30}$$

$$= \mathcal{L}(\mathcal{D}; \mathbf{w}_t) - \eta\left(1 - \xi_t - \frac{\beta\eta(1-\xi_t)}{2} + \frac{\beta\eta\xi_t}{2} + \frac{\beta^3\eta\rho^2\xi_t}{2} + (1-\beta\eta)(1-\beta\rho)\xi_t\right)\|\mathbf{g}_t\|^2 \tag{31}$$

$$= \mathcal{L}(\mathcal{D}; \mathbf{w}_t) - \eta\left(1 - \frac{\beta\eta(1-\xi_t)}{2} + \frac{\beta\eta\xi_t}{2} + \frac{\beta^3\eta\xi_t\rho^2}{2} - \beta\eta\xi_t - \beta\rho\xi_t + \beta^2\eta\rho\xi_t\right)\|\mathbf{g}_t\|^2$$

$$\leq \mathcal{L}(\mathcal{D}; \mathbf{w}_t) - \eta\left(1 - \frac{\beta\eta}{2} - \beta\rho\xi_t\right)\|\mathbf{g}_t\|^2, \tag{32}$$

where we have used $\|\mathbf{h}_t - \mathbf{g}_t\|^2 = \|\nabla\mathcal{L}(\mathcal{D}; \mathbf{w}_t + \rho\nabla\mathcal{L}(\mathcal{D}; \mathbf{w}_t)) - \nabla\mathcal{L}(\mathcal{D}; \mathbf{w}_t)\|^2 \leq \beta^2\rho^2\|\nabla\mathcal{L}(\mathcal{D}; \mathbf{w}_t)\|^2 = \beta^2\rho^2\|\mathbf{g}_t\|^2$ to obtain (30), and Lemma A.5 to obtain (31).

Summing over $t$ from $t = 0$ to $T - 1$ on both sides of (32) and rearranging, we have

$$\sum_{t=0}^{T-1}\eta\left(1 - \frac{\beta\eta}{2} - \beta\rho\xi_t\right)\|\mathbf{g}_t\|^2 \leq \mathcal{L}(\mathcal{D}; \mathbf{w}_0) - \mathcal{L}(\mathcal{D}; \mathbf{w}_T). \tag{33}$$

As $\rho < \frac{1}{2\beta}$ and $\eta < \frac{1}{\beta}$, it follows that $1 - \frac{\beta\eta}{2} - \beta\rho\xi_t > 0$ for all $t$. Thus, (33) implies

$$\min_{0\leq t\leq T-1}\|\mathbf{g}_t\|^2 \leq \frac{\mathcal{L}(\mathcal{D}; \mathbf{w}_0) - \mathcal{L}(\mathcal{D}; \mathbf{w}_T)}{\sum_{t=0}^{T-1}\eta\left(1 - \frac{\beta\eta}{2} - \xi_t\beta\rho\right)} = \frac{\mathcal{L}(\mathcal{D}; \mathbf{w}_0) - \mathcal{L}(\mathcal{D}; \mathbf{w}_T)}{T\eta\left(1 - \frac{\beta\eta}{2} - \beta\rho\zeta\right)}, \tag{34}$$

where $\zeta = \frac{1}{T}\sum_{t=0}^{T-1}\xi_t \in [0, 1]$ and we finish the proof. $\square$

# B   ADDITIONAL EXPERIMENTAL RESULTS

## B.1   DISTRIBUTION OF STOCHASTIC GRADIENT NORMS

Figure 8 shows the distributions of stochastic gradient norms for *ResNet-18*, *WRN-28-10* and *PyramidNet-110* on *CIFAR-10* and *CIFAR-100*. As can be seen, the distribution follows a Bell curve in all settings. Figure 9 shows the Q-Q plots. We can see that the curves are close to the lines.

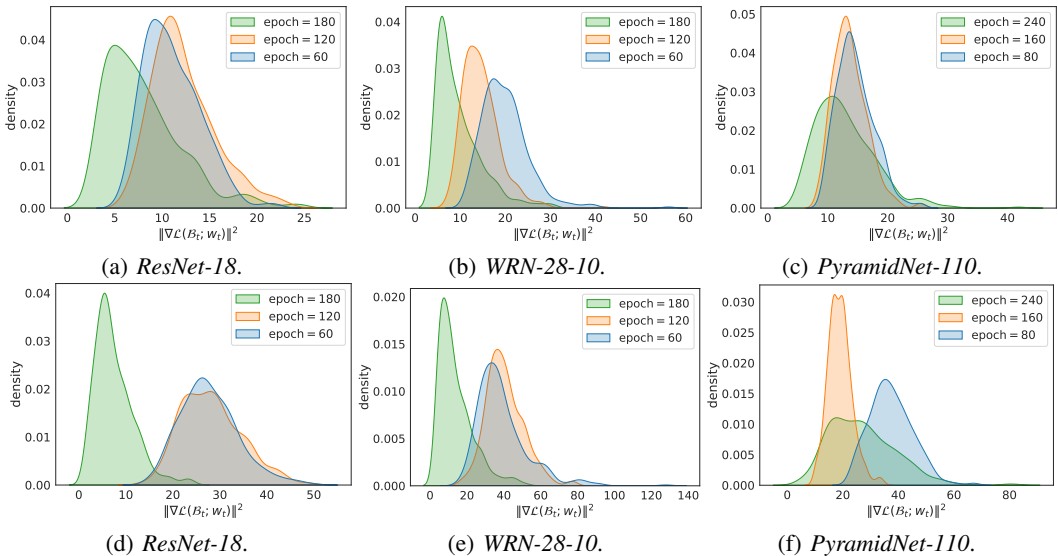

Figure 8: Distributions of stochastic gradient norms on *CIFAR-10* (top) and *CIFAR-100* (bottom). Best viewed in color.

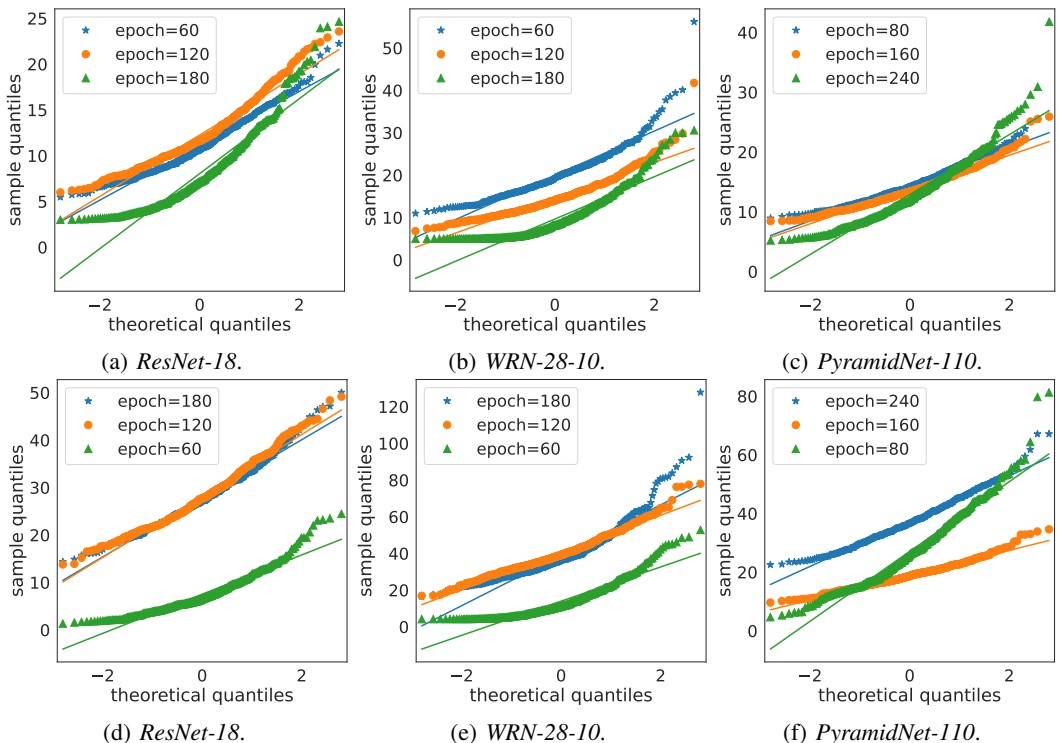

Figure 9: Q-Q plots of stochastic gradient norms on *CIFAR-10* (top) and *CIFAR-100* (bottom). Best viewed in color.

B.2 EFFECT OF $k$ ON LOOKSAM

In this experiment, we demonstrate that LookSAM is sensitive to the choice of $k$. Table 4 shows the testing accuracy and fraction of SAM updates when using LookSAM on noisy *CIFAR-10*, with $k \in \{2, 3, 4, 5\}$ and the *ResNet-18* model. As can be seen, $k = 2$ yields much better performance than $k \in \{3, 4, 5\}$, particularly at higher noise levels (e.g., $80\%$).

Table 4: Effects of $k$ in LookSAM on *CIFAR-10* with different levels of label noise using *ResNet-18*.

| | noise $= 20\%$ | | noise $= 40\%$ | | noise $= 60\%$ | | noise $= 80\%$ | |
|---|---|---|---|---|---|---|---|---|
| $k$ | accuracy | %SAM | accuracy | %SAM | accuracy | %SAM | accuracy | %SAM |
| 2 | **92.72** | 50.0 | **88.04** | 50.0 | **72.26** | 50.0 | **69.72** | 50.0 |
| 3 | 89.07 | 33.3 | 75.38 | 33.3 | 63.79 | 33.3 | 53.87 | 33.3 |
| 4 | 89.00 | 25.0 | 74.12 | 25.0 | 58.17 | 25.0 | 52.28 | 25.0 |
| 5 | 88.57 | 20.0 | 73.90 | 20.0 | 56.80 | 20.0 | 51.82 | 20.0 |

### B.3 MORE RESULTS ON ROBUSTNESS TO LABEL NOISE

Figure 10 (resp. 11) shows the curves of accuracies at noise levels of $20\%$, $40\%$, $60\%$, and $80\%$ with *ResNet-18* (resp. *ResNet-32*). As can be seen, in all settings, AE-LookSAM is as robust to label noise as SAM.

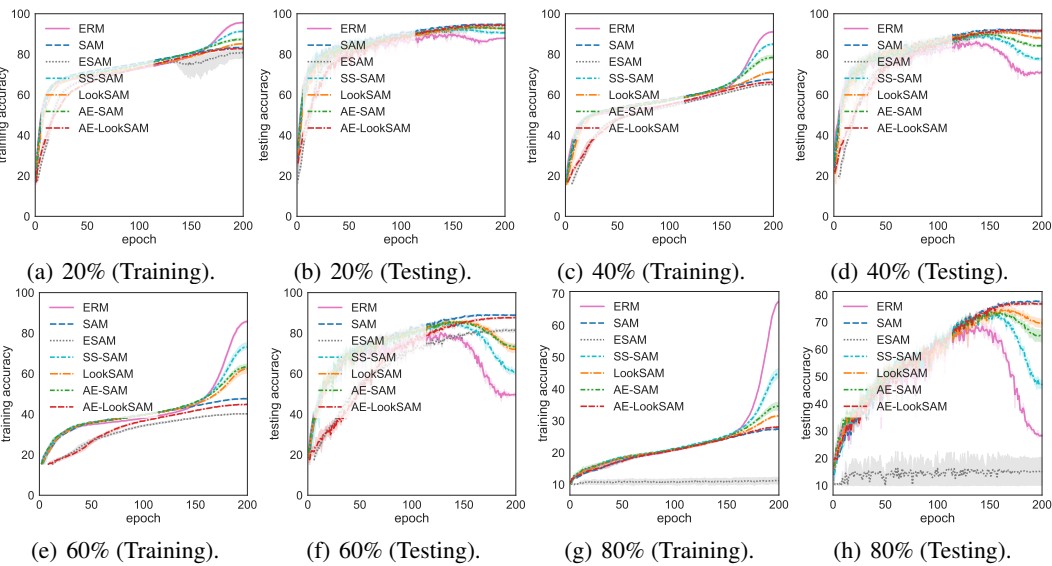

(a) 20% (Training).    (b) 20% (Testing).    (c) 40% (Training).    (d) 40% (Testing).

(e) 60% (Training).    (f) 60% (Testing).    (g) 80% (Training).    (h) 80% (Testing).

Figure 10: Accuracies with number of epochs on CIFAR-10 with $20\%$, $40\%$, $60\%$, and $80\%$ noise level using *ResNet-18*. Best viewed in color.

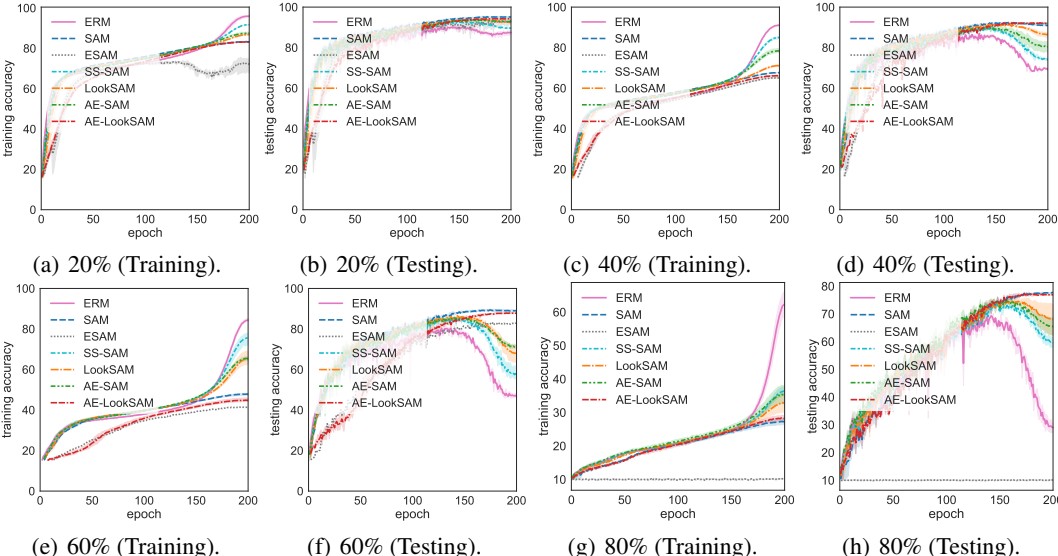

(a) 20% (Training).    (b) 20% (Testing).    (c) 40% (Training).    (d) 40% (Testing).

(e) 60% (Training).    (f) 60% (Testing).    (g) 80% (Training).    (h) 80% (Testing).

Figure 11: Accuracies with number of epochs on CIFAR-10 with $20\%$, $40\%$, $60\%$, and $80\%$ noise level using *ResNet-32*. Best viewed in color.

### B.4 EFFECTS OF $\lambda_1$ AND $\lambda_2$ ON AE-LOOKSAM

In this experiment, we study the effects of $\lambda_1$ and $\lambda_2$ on AE-LookSAM. Experiment is performed on *CIFAR-10* with label noise ($80\%$ noisy labels), using the same setup as in Section 4.3.

Figure 12 shows the effects of $\lambda_1$ and $\lambda_2$ on the fraction of SAM updates. Again, as in Section 4.4, for a fixed $\lambda_2$, increasing $\lambda_1$ always reduces the fraction of SAM updates. Figure 13 shows the effects of $\lambda_1$ and $\lambda_2$ on the testing accuracy of AE-SAM. As can be seen, the observations are similar to those in Section 4.4.

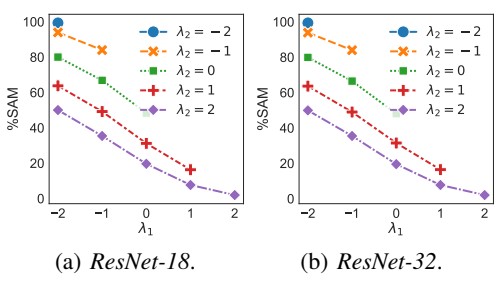

| (a) *ResNet-18*. | (b) *ResNet-32*. | (a) *ResNet-18*. | (b) *ResNet-32*. |

Figure 12: Effects of $\lambda_1$ and $\lambda_2$ on fraction of SAM updates on *CIFAR-10* (with $80\%$ noisy labels). Best viewed in color.

Figure 13: Effects of $\lambda_1$ and $\lambda_2$ on testing accuracy of *CIFAR-10* (with $80\%$ noisy labels). Note that the curves for $\lambda_2 \in \{-2, -1\}$ overlap completely with that of $\lambda_2 = 1$. Best viewed in color.

### B.5 ADDITIONAL CONVERGENCE RESULTS ON *CIFAR-10* AND *CIFAR-100*

Figure 14 shows convergence of AE-SAM's training loss on the *CIFAR-10* and *CIFAR-100* datasets. As can be seen, AE-SAM achieves convergence with various network architectures.

Figure 15 shows the training losses w.r.t. the number of epochs for AE-SAM and SS-SAM. As can be seen, AE-SAM and SS-SAM converge with comparable speeds, which agrees with Theorem 3.3 as both of them have comparable fractions of SAM updates (Table 1).

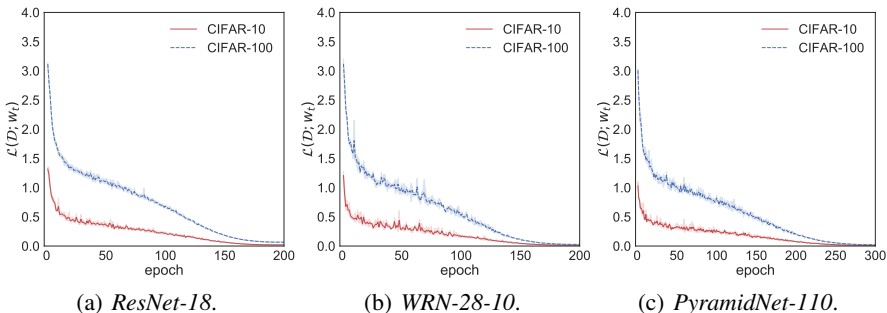

| (a) *ResNet-18*. | (b) *WRN-28-10*. | (c) *PyramidNet-110*. |

Figure 14: Training loss of AE-SAM with number of epochs on *CIFAR-10* and *CIFAR-100*. Best viewed in color.

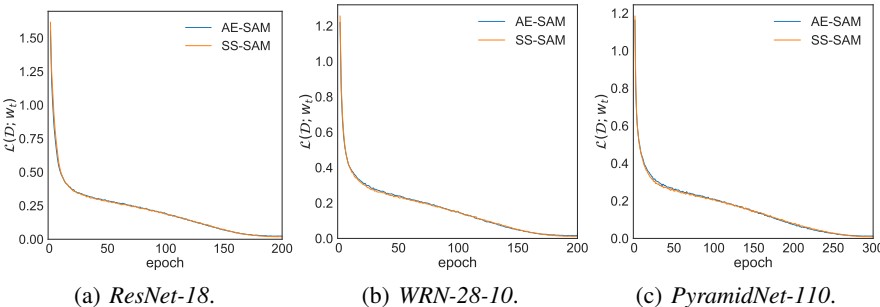

| (a) *ResNet-18*. | (b) *WRN-28-10*. | (c) *PyramidNet-110*. |

Figure 15: Training losses of AE-SAM and SS-SAM with number of epochs on *CIFAR-10*. Note that the two curves almost completely overlap. Best viewed in color.

