# OpenReview forum: "An Adaptive Policy to Employ Sharpness-Aware Minimization"
_ICLR.cc/2023/Conference — ICLR 2023 poster_

### Official Review · Reviewer_4reb · 2022-10-23

**Confidence:** 3
**Correctness:** 3
**Technical Novelty And Significance:** 2
**Empirical Novelty And Significance:** 2
**Recommendation:** 6

**Clarity, Quality, Novelty And Reproducibility:**

As I mentioned in the "weakness" in the last section, the author is suggested to clarify how and why they choose the formulation in (4) and (5) for adaptive SAM.

**Strength And Weaknesses:**

Strength:
(1) The proposed method is new and intuitive
(2) The effectiveness of proposed algorithm is supported by both theoretical and empirical evidence


Weakness:
(1) It is not clear how the author come up with (4) (5) for mean and variance derivation. It seems that this formulation enables establishing convergence result but it is hard to see the intuition behind this formulation.

**Summary Of The Paper:**

This paper proposed an adapt scheme for sharpness-aware minimization (SAM), in which the algorithm simply perform ERM when landscape is flat and perform SAM which landscape is sharp. This method is based on couple of approximation strategies. The author also establish the theoretical guarantee for their proposed algorithm. Some empirical results are provided to verify the effectiveness of the proposed algorithm.

**Summary Of The Review:**

Overall this paper is well-written and easy to follow. The high level idea of the algorithm proposed in this paper is intuitive and nature and should be interesting to the community. Since i didn't see obvious weakness in both theoretical and empirical results, I will give the score of 6 for now.

---

> ### Author Response · Authors · 2022-11-17
> **Reply to Reviewer 4reb**
>
> Thank you for your thoughtful review
> and valuable feedback.
> We address your concerns as follows.
>
> ---
>
> > Q1.
> "It is not clear how the author come up with (4) (5) for mean and variance derivation",
> "the author is suggested to clarify how and why they choose the formulation in (4) and (5) for adaptive SAM."
>
> **A1.**
> The exponential moving average (EMA)
> used in (4) and (5)
> is popularly used
> in adaptive gradient methods (e.g.,
> RMSProp [1],
> AdaDelta [2],
> AdaBelief [3],
> Adam [4]).
> For example,
> in the Adam optimizer,
> its first and second moments are estimated by EMA.
> Compared with the arithmetic mean (i.e., sampling a large number of
> mini-batches and estimate the first and second moments by arithmetic mean),
> EMA is much **cheaper**.
> Hence, we use EMA for estimating the mean and variance of the stochastic gradient norm.
>
> ---
>
> > Q2. "It seems that this formulation
> enables establishing convergence result but it is hard to see the intuition behind this formulation."
>
> **A2.**
> Note that our Theorem 3.3 (in the revised version)
> holds for **any** algorithm that mixes SAM and ERM.
> As
> the EMA formulation only affects  the mixing rule
> (Step 6 of Algorithm 1),
> it is not a condition for establishing convergence.
> As mentioned in the above reply,
> the intuition of EMA is from popular adaptive gradient methods,
> and
> we use EMA for computational efficiency.
>
> ---
>
> **References**
>
> [1] Tijmen Tieleman and Geoffrey Hinton. Lecture 6.5-RMSProp, neural networks for machine learning. Technical report, 2012. \
> [2] Matthew D Zeiler. AdaDelta: an adaptive learning rate method. Preprint arXiv:1212.5701, 2012. \
> [3] Juntang Zhuang, Tommy Tang, Yifan Ding, Sekhar C Tatikonda, Nicha Dvornek, Xenophon Papademetris, and James Duncan. AdaBelief optimizer: Adapting stepsizes by the belief in observed gradients. In _Neural Information Processing Systems_, 2020. \
> [4] Diederik P Kingma and Jimmy Ba. Adam: A method for stochastic optimization. In _International Conference on Learning Representations_, 2015.

---

> > ### Comment · Reviewer_4reb · 2022-12-14
> > **Thanks for reply**
> >
> > Thank for author's detailed explaination. I am now very clear about the intuition behind the design of the proposed algorithm. Thus, I will keep my score as weakly accept.

---

### Official Review · Reviewer_rg2h · 2022-10-23

**Confidence:** 5
**Clarity, Quality, Novelty And Reproducibility:** This paper is well written. However, …
**Correctness:** 3
**Technical Novelty And Significance:** 2
**Empirical Novelty And Significance:** 2
**Recommendation:** 5

**Strength And Weaknesses:**

The authors propose a new variant of the SAM optimizer. Preliminary experiments demonstrate its effectiveness. However, before this work is accepted, several concerns should be resolved:
(i)	The proposed AE-SAM falls into the scope of mixing SGD and SAM. Several existing works adopt a similar idea to improve the efficacy of SAM, including SS-SAM, and Gradient penalized SAM (Zhao et al, ICML 2022). The idea is not new, which limits the novelty of the proposed approach.
(ii)	About the practicality. Several hyperparameters are introduced in AE-SAM. In general, SAM-type methods are sensitive with respect to the perturbation parameters. More hyperparameters may make the tuning process more difficult.
(iii)	About the theory. In general, mini-batch SGD could achieve the linear speedup property with respect to the mini-batch size. (see Efficient Mini-batch Training for Stochastic Optimization, KDD). However, in Theorem 3.4, the linear speedup property is missing. In addition, can the author prove that the proposed AE-SAM theoretically outperforms randomized SAM (SS-SAM)?
(iv)	About the sensitivity of the adaptive policy. The proposed adaptive policy is highly related to mini-batch size. When training a deep neural network with a different mini-batch size, it shows different gradient variances. (see Not All Layers Are Equal: A Layer-Wise Adaptive Approach Toward Large-Scale DNN Training, WWW2022). The authors may conduct more experiments to show the consistency/sensitivity of the adaptive policy with respect to mini-batch size.
(v)	More experiments verification on advanced neural network architecture are needed, such as vision transformer (such as Deit).



**Summary Of The Paper:**

In this work, the authors propose a new variant of SAM optimizer, called AE-SAM by mixing SAM and SGD with a carefully designed condition. Moreover, the theoretical convergence of the proposed algorithm is also provided. Preliminary experiments demonstrate the efficacy of the proposed approach.

**Summary Of The Review:**

See the comments above.

---

> ### Author Response · Authors · 2022-11-17
> **Reply to Reviewer rg2h (3/3)**
>
> > Q7. "More experiments verification on advanced neural network architecture are needed, such as vision transformer (such as Deit)."
>
> **A7.**
> As suggested,
> we conducted experiments on vision transformer using the setting in
> [1] (which is mentioned by the reviewer).
> Table below
> shows the testing accuracy.
> As can be seen,
> AE-SAM performs better than SAM while using only $50\\%$
> of SAM updates.
> SS-SAM and AE-SAM have comparable
> %SAM (about $50\\%$),
> and AE-SAM achieves higher accuracy.
> As for LookSAM and AE-LookSAM,
> both use about $20\\%$ of SAM updates but AE-LookSAM is better.
> These improvements confirm the effectiveness of the adaptive policy
> for *ViT-S16*.
>
> |         |       &emsp;&emsp; *CIFAR-10*           |    |&emsp;&emsp; *CIFAR-100* |   |
> | :-------: | :-----------: | :-----------: | :-----------: | :-----------: |
> |       | Accuracy    | %SAM        | Accuracy | %SAM |
> | ERM      | $86.69 \pm0.11$ | $0.0 \pm0.0$ | $62.42 \pm0.22$ | $0.0 \pm0.0$ |
> |      |   |  |   |   |
> |      |   |  |   |   |
> | SAM      | $87.37 \pm0.09$ | $100.0 \pm0.0$ | $63.23 \pm0.22$ | $100.0 \pm0.0$ |
> | ESAM      | $84.27 \pm0.11$ | $100.0 \pm0.0$ | $62.11 \pm0.15$ | $100.0 \pm0.0$ |
> | Gradient penalized SAM [1]     | $87.71 \pm0.19$ | $100.0 \pm0.0$ | $63.41 \pm0.22$ | $100.0 \pm0.0$ |
> |      |   |  |   |   |
> |      |   |  |   |   |
> | SS-SAM      | $87.38 \pm0.14$ | $50.0 \pm0.0$ | $63.18 \pm0.19$ | $50.0 \pm0.0$ |
> | AE-SAM      | $\underline{\mathbf{87.77}} \pm0.13$ | $49.7 \pm0.1$ | $\underline{63.68} \pm0.23$ | $49.5 \pm0.2$ |
> |     |   |  |   |   |
> |      |   |  |   |   |
> | LookSAM      | $87.12 \pm0.20$ | $20.0 \pm0.0$ | $63.52 \pm0.19$ | $20.0 \pm0.0$ |
> | AE-LookSAM      | $\underline{87.32} \pm0.11$ | $20.2 \pm0.2$ | $\underline{\mathbf{64.16}} \pm0.23 $| $20.3 \pm0.2$ |
>
> (The highest accuracy in each
> group is underlined; while the highest accuracy (across all groups) is in
> bold.)
>
>
> ---
>
> **References**
>
> [1] Yang Zhao, Hao Zhang, and Xiuyuan Hu. Penalizing gradient norm for efficiently improving
> generalization in deep learning. In _International Conference on Machine Learning_, 2022a. \
> [2] Yang Zhao, Hao Zhang, and Xiuyuan Hu. SS-SAM: Stochastic scheduled sharpness-aware minimization
> for efficiently training deep neural networks. Preprint arXiv:2203.09962, 2022b. \
> [3] Yong Liu, Siqi Mai, Xiangning Chen, Cho-Jui Hsieh, and Yang You. Towards efficient and scalable
> sharpness-aware minimization. In _IEEE Conference on Computer Vision and Pattern Recognition_, 2022. \
> [4] Mu Li, Tong Zhang, Yuqiang Chen, and Alexander J Smola. Efficient mini-batch training for
> stochastic optimization. In _ACM SIGKDD International Conference on Knowledge Discovery and
> Data Mining_, 2014. \
> [5] Yunyong Ko, Dongwon Lee, and Sang-Wook Kim. Not all layers are equal: A layer-wise adaptive
> approach toward large-scale DNN training. In _ACM Web Conference_, 2022.

---

> ### Author Response · Authors · 2022-11-17
> **Reply to Reviewer rg2h (2/3)**
>
> > Q4. "About the theory.
> In general, mini-batch SGD could achieve the linear speedup property with respect to the mini-batch size. (see Efficient Mini-batch Training for Stochastic Optimization, KDD).
> However, in Theorem 3.4, the linear speedup property is missing."
>
> **A4.**
> First, note that
> in the  above
> paper
> mentioned  by the reviewer,
> they only claim a sublinear rate
> (Theorem 1, Section 2.3,
> [4]),
> not
> linear speedup.
> Moreover, while
> their algorithm
> achieves a $\mathcal{O}\left(1/\sqrt{Tb}\right)$ rate
> (where $b$ is the batch size and $T$ is the number of iterations),
> this holds only
> for **convex** losses and also
> by
> solving a regularized minimization problem
> **exactly** at every iteration.
> However,
> in modern deep learning, the loss is typically nonconvex and
> seeking an exact minimizer in each iteration is also computationally infeasible.
>
> Besides, we can also
> achieve a $\mathcal{O}\left(1/\sqrt{Tb}\right)$ rate
> if $\eta = \frac{\sqrt{b}}{4\beta \sqrt{T}}$ and $T\geq b$.
> Specifically,
> \begin{align}
>     \min_{0\leq t\leq T-1} \mathbb{E}  \Vert \nabla \mathcal{L}(\mathcal{D}; {\bf w}_t) \Vert^2
>     \leq
>     \frac{32\beta(\mathbb{E} \\mathcal{L}(\mathcal{D}; {\bf w}_0) - \mathbb{E} \mathcal{L}(\mathcal{D}; {\bf w}_T))}{\sqrt{Tb}(7-6\zeta )}
>     + \frac{(1 +  \zeta + 5\beta^2   \zeta )\sigma^2}{\sqrt{Tb}(7-6\zeta)},
> \end{align}
> where
> $\zeta\in[0,1]$.
> Note that unlike
> [4], our result holds for nonconvex loss and does NOT
> require finding an exact minimizer in each iteration.
> This result is
> now added
> to Corollary A.3 of Appendix A in this revised version.
> Moreover,
> note that
> the regularization hyperparameter $\gamma$
> in [4]
> is chosen as
> $\mathcal{O}(\sqrt{T/b})$,
> which plays the same role as $\eta=\mathcal{O}(\sqrt{b/T})$ here.
>
> ---
>
> > Q5. "can the author prove that the proposed AE-SAM theoretically outperforms randomized SAM (SS-SAM)?"
>
> **A5.**
> Convergence analysis for
> SS-SAM is not provided in [2].
> However, as mentioned in our reply to Q2 of
> reviewer qUYU,
> our
> Theorem 3.3
> holds for any algorithm that mixes SAM and ERM (and thus includes
> SS-SAM).
> We can see from the theorem that
> the upper bound of the convergence rate
> depends on the fraction of SAM updates.
> As AE-SAM and SS-SAM have
> comparable fractions empirically (Table 1),
> their convergence rates are also comparable,
> which is empirically verified by the newly added
> Figure 21 in Appendix B.6.
> For generalization performance,
> empirical results in Tables 1,2,3 demonstrate that AE-SAM performs better.
>
> > Q6. "The proposed adaptive policy is highly related to mini-batch
> size.
> When training a deep neural network with a different mini-batch size,it shows different gradient variances. (see Not All Layers Are Equal: A Layer-Wise Adaptive Approach Toward Large-Scale DNN Training, WWW2022).
> The authors may conduct more experiments to show the consistency/sensitivity of the adaptive policy with respect to mini-batch size."
>
> **A6.**
> As suggested,
> we conducted experiments to study the effect of mini-batch size on AE-SAM.
> Figure 20 in Appendix B.4
> shows the curves of
> gradient variance on *CIFAR-10* and *CIFAR-100*
> with different batch sizes.
> As we can see,
> variances are related to batch sizes,
> which is consistent with the observation
> in the mentioned paper [5].
>
> Table below
> reports the testing accuracy of AE-SAM with different batch sizes.
> As can be seen,
> AE-SAM is insensitive over a large range of batch size.
> In particular, $b=128$ is a good choice and is used in
> the experiments.
>
> |         |       &emsp;&emsp; *CIFAR-10*           |    |&emsp;&emsp; *CIFAR-100* |   |
> | :-------: | :-----------: | :-----------: | :-----------: | :-----------: |
> |    $b$   | Accuracy    | %SAM        | Accuracy | %SAM |
> | 16      | $94.99 \pm0.22$ | $49.3 \pm0.2$ |$ 77.40 \pm0.56$ | $49.2 \pm0.2$ |
> | 32      | $95.98 \pm0.11 $|$ 49.3 \pm0.2 $| $79.88 \pm0.13 $|$ 49.2 \pm0.1 $|
> | 64      | $96.57 \pm0.10$ |$ 49.4 \pm0.2$ |$ 80.23 \pm0.30$ | $49.5 \pm0.0 $|
> | 128      | $\mathbf{96.63} \pm0.04$ |$ 50.1 \pm0.1 $| $\mathbf{80.48} \pm0.11$ |$ 49.8 \pm0.0 $|
> | 256      | $96.47 \pm0.11 $| $49.1 \pm0.1 $|$79.71 \pm0.13$| $50.2 \pm0.2 $|
> | 512      | $95.93 \pm0.08$ |$ 49.8 \pm0.1$ |$ 78.63 \pm0.29 $|$ 49.9 \pm0.2$ |
> | 1024    | $95.08 \pm0.25$ |$ 50.0 \pm0.3 $|$ 77.18 \pm0.13 $|$ 50.1 \pm0.1 $|
> | 2048    | $94.24 \pm0.48$ |$ 49.3 \pm0.0 $|$ 75.78 \pm0.18 $|$ 49.7 \pm0.1 $|
> | 4096    | $91.35 \pm0.74$ |$ 49.6 \pm0.2 $|$ 72.59 \pm0.32 $|$ 49.5 \pm0.2$ |
> | 8192    | $89.19 \pm1.71 $|$ 50.1 \pm0.1 $|$ 67.83 \pm0.64 $|$ 49.6 \pm0.3 $|

---

> ### Author Response · Authors · 2022-11-17
> **Reply to Reviewer rg2h (1/3)**
>
> Thank you for your thoughtful review and valuable feedback.
> We address your concerns as follows.
>
> ---
>
> > Q1.
> "The proposed AE-SAM falls into the scope of mixing SGD and SAM.
> Several existing works
> adopt a similar idea to improve the efficacy of SAM,
> including SS-SAM, and Gradient penalized SAM (Zhao et al, ICML 2022).
> The idea is not new, which limits the novelty of the proposed approach.",
> "However, mixing SGD and SAM is not new."
>
> **A1.**
> The following three works are on
> mixing SAM and SGD.
> However, they all have some limitations:
> (i)
> Gradient penalized SAM [1]:
> it mixes SAM and SGD
> at every iteration.
> However, it
> suffers from having the same computation burden as SAM;
> (ii) SS-SAM [2], which
> mixes SAM and SGD randomly;
> (iii)
> LookSAM [3], which
> adopts a periodic policy.
> While both SS-SAM and LookSAM
> improve the efficiency by reducing the fraction of SAM updates,
> unlike the proposed AE-SAM,
> their mixing policies are
> not adaptive to the loss landscape.
> Intuitively, the
> SAM update is more suitable for sharp regions than for flat regions.
>
>
> The adaptive policy is a novel way to mix
> SAM and ERM,
> as also mentioned by the other two reviewers:
> "Adaptive thresholding is a **new** approach to mixing SAM and SGD/ERM"
> (reviewer qUYU);
> "The proposed method is **new** and intuitive"
> (reviewer 4reb).
>
> ---
>
> > Q2. "Several hyperparameters are introduced in AE-SAM''
> > and ``More hyperparameters may make the tuning process more difficult."
>
> **A2.**
> Compared with SAM,
> AE-SAM has three additional hyperparameters:
> (i) $\delta$ in exponential
> moving average (Eq.(4) and (5)).
> This is always set to $0.9$
> in the experiments;
> (ii) $\lambda_1$ and $\lambda_2$ in
> the schedule $c_t = g_{\lambda_1, \lambda_2}(t)$.
> Ablation study in Section 4.4
> demonstrates the effects of $(\lambda_1, \lambda_2)$ on the testing accuracy of
> AE-SAM.
> Specifically,
> in the experiments,
> $\lambda_1$ and $\lambda_2$
> are simply fixed at $-1$ and $1$, respectively.
> Hence,
> the newly introduced hyperparameters can all be fixed and do NOT have to be
> tuned.
>
> ---
>
> > Q3. "In general,
> SAM-type methods are sensitive with respect to the perturbation
> parameters."
>
> **A3.**
> Table below
> reports the effects of perturbance radius $\rho$
> on the testing accuracy of AE-SAM.
> As can be seen,
> AE-SAM is insensitive over a large range of $\rho$ (e.g.,
> $\rho \in \{0.05, 0.1\}$).
> In the experiments,
> $\rho=0.05$ and $0.1$ are chosen for *CIFAR-10* and *CIFAR-100*,
> respectively,
> as is common in the SAM literature (e.g., SAM and ESAM).
>
> |         |       &emsp;&emsp; *CIFAR-10*           |    |&emsp;&emsp; *CIFAR-100* |   |
> | -------: | :-----------: | :-----------: | :-----------: | :-----------: |
> |  $\rho$     | Accuracy    | %SAM        | Accuracy | %SAM |
> | 0.00001 | $95.82 \pm0.26 $|$ 49.7 \pm0.1 $|$ 79.23 \pm0.31 $|$ 49.9 \pm0.0 $|$
> | 0.0001   | $96.03 \pm0.16 $|$ 50.1 \pm0.2 $|$ 79.02 \pm0.20 $|$ 50.1 \pm0.1 $|
> | 0.001     | $96.21 \pm0.03 $|$ 50.0 \pm0.0 $|$ 79.36 \pm0.19 $|$ 49.8 \pm0.0 $|
> | 0.01       | $96.31 \pm0.04 $|$ 49.8 \pm0.1 $|$ 79.55 \pm0.08 $|$ 49.7 \pm0.0 $|
> | 0.02       | $96.44 \pm0.03 $|$ 50.0 \pm0.1 $|$ 79.72 \pm0.54 $|$ 49.8 \pm0.0 $|
> | 0.05       | $\mathbf{96.63} \pm0.04 $|$ 50.1 \pm0.1 $|$ 80.37 \pm0.19 $|$ 49.9 \pm0.0 $|
> | 0.1         | $96.53 \pm0.05 $|$ 50.0 \pm0.1 $|$\mathbf{80.48} \pm0.11 $|$ 49.8 \pm0.0 $|
> | 0.2         | $96.45 \pm0.17 $|$ 49.8 \pm0.1 $|$ 80.46 \pm0.26 $|$ 49.9 \pm0.0 $|
> | 0.5         | $95.52 \pm0.20 $|$ 49.6 \pm0.2 $|$ 80.28 \pm0.22 $|$ 49.8 \pm0.1 $|
> | 1.0         | $93.98 \pm0.19 $|$ 49.8 \pm0.0 $|$ 80.11 \pm0.06 $|$ 50.0 \pm0.1 $|
> | 2.0         | $91.71 \pm0.21 $|$ 50.3 \pm0.2 $|$ 78.56 \pm0.19 $|$ 50.1 \pm0.2 $|
> | 5.0         | $83.19 \pm0.67 $|$ 50.1 \pm0.1 $|$ 71.37 \pm0.62 $|$ 49.3 \pm0.2 $|

---

> ### Comment · Reviewer_rg2h · 2022-12-10
> **Thanks for your response.**
>
> We thank the reviewers' insightful comments.  Most of my concerns have been solved. I will increase my initial score to borderline reject (score 5). In general,  the authors speed great effort to improve the efficacy of SAM by adopting a mixing technique, which has been partially proposed by recent works.  Thus, the main contribution is still limited.
>
> Overall, based on the current version, this work is still a borderline paper and I lean to reject it.

---

> > ### Author Response · Authors · 2022-12-10
> > **Reply to the further comments**
> >
> > Thank you for increasing the score! We sincerely appreciate your further comments! We would like to address your remaining concern as follows.
> >
> > ---
> >
> > > **Q**. “In general, the authors speed great effort to improve the efficacy of SAM by adopting a mixing technique, which has been partially proposed by recent works. Thus, the main contribution is still limited.”
> >
> > **A.** The proposed mixing policy differs from existing mixing methods and can overcome their limitations: (i) Gradient penalized SAM [1] mixes SAM and ERM at **every** iteration, which is computationally intensive. The proposed AE-SAM switches between SAM and ERM adaptively, reducing the fraction of SAM updates and improving efficiency. (ii) SS-SAM [2] mixes SAM and ERM randomly; (iii) LookSAM [3] adopts a periodic policy. Intuitively, the SAM update is more suitable for sharp regions than for flat regions. However, the mixing policies in SS-SAM and LookSAM are **not adaptive to loss landscape**.
> >
> > In our paper, we design an **adaptive** policy to employ SAM based on the loss landscape geometry: The SAM update is used when the model is in sharp regions, while the ERM update is used in flat regions. Compared with random or periodic policies, the adaptive policy is a **novel** way to mix SAM and ERM, as acknowledged by the other two reviewers.  Furthermore, empirical results (Table 1) show that the adaptive policy performs better, demonstrating its **effectiveness**.

---

### Official Review · Reviewer_qUYU · 2022-10-26

**Confidence:** 4
**Clarity, Quality, Novelty And Reproducibility:** The paper is well written and easy to…
**Correctness:** 3
**Technical Novelty And Significance:** 2
**Empirical Novelty And Significance:** Not applicable
**Recommendation:** 6

**Strength And Weaknesses:**

Strength: Adaptive thresholding is a new approach to mixing SAM and SGD/ERM
Weaknesses: choosing this threshold is not well justified.

**Summary Of The Paper:**

Sharpness-aware minimization (SAM), which searches for flat minima by min-max optimization, has been shown to be useful in improving model generalization. However, SAM requires two gradient evaluations per iteration which makes it cost twice more expensive as SGD. One way to reduce this cost is mixing SAM with SGD in an adaptive manner. This paper proposes an adaptation based on the stochastic gradient’s norm.  The main idea is to keep an adaptive threshold (updating it per iterate), and then compare the gradient norm with this threshold: if it is bigger take a SAM update, otherwise take a SGD update.  They show the convergence of their proposed method.


**Summary Of The Review:**

Comments:
1- Assuming that the stochastic gradient follows a normal distribution is not well justified. For example, there exists research in the literature that shows this norm follows heavy-tailed distribution and not normal distribution.

2- In your analysis, the adaptivity of the threshold doesn’t show up and the analysis is very similar to the method which takes SAM step randomly. Specifically, in the deterministic setting, there is no stochastic gradient and hence normal distribution assumption is incorrect. Besides in the empirical result, there is no significant difference between AE-SAM and SS-SAM.

3- It is not explained in the paper why using AE-SAM can improve the robustness against label noises.

---

> ### Author Response · Authors · 2022-11-17
> **Reply to Reviewer qUYU (2/2)**
>
> > Q5.
> "It is not explained in the paper
> why using AE-SAM can improve the robustness against label noises."
>
> **A5.**
> [8-10]
> have empirically observed that
> SAM is robust to label noise.
> However,
> none of them
> can provide theoretical
> explanations
> on why SAM can improve robustness.
> Thus, this is still an **open question** for
> SAM-based methods (including AE-SAM).
>
> ---
>
> **References**
>
> [1] Umut Simsekli, Levent Sagun, and Mert Gurbuzbalaban. A tail-index analysis of stochastic gradient
> noise in deep neural networks. In _International Conference on Machine Learning_, 2019. \
> [2] Thanh Huy Nguyen, Umut Simsekli, Mert Gurbuzbalaban, and Gael Richard. First exit time analysis
> of stochastic gradient descent under heavy-tailed gradient noise. In _Neural Information Processing
> Systems_, 2019. \
> [3] Zeke Xie, Issei Sato, and Masashi Sugiyama. A diffusion theory for deep learning dynamics:
> Stochastic gradient descent exponentially favors flat minima. In _International Conference on
> Learning Representations_, 2021. \
> [4] Stephan Mandt, Matthew D Hoffman, and David M Blei. Stochastic gradient descent as approximate
> bayesian inference. _Journal of Machine Learning Research_, 18:1–35, 2017. \
> [5] Zhanxing Zhu, Jingfeng Wu, Bing Yu, Lei Wu, and Jinwen Ma. The anisotropic noise in stochastic
> gradient descent: Its behavior of escaping from sharp minima and regularization effects. In
> _International Conference on Machine Learning_, 2019. \
> [6] Yang Zhao, Hao Zhang, and Xiuyuan Hu. SS-SAM: Stochastic scheduled sharpness-aware minimization
> for efficiently training deep neural networks. Preprint arXiv:2203.09962, 2022b. \
> [7] Yong Liu, Siqi Mai, Xiangning Chen, Cho-Jui Hsieh, and Yang You. Towards efficient and scalable
> sharpness-aware minimization. In _IEEE Conference on Computer Vision and Pattern Recognition_, 2022. \
> [8] Pierre Foret, Ariel Kleiner, Hossein Mobahi, and Behnam Neyshabur. Sharpness-aware minimization
> for efficiently improving generalization. In _International Conference on Learning Representations_, 2021. \
> [9] Jungmin Kwon, Jeongseop Kim, Hyunseo Park, and In Kwon Choi. ASAM: Adaptive sharpness aware
> minimization for scale-invariant learning of deep neural networks. In _International Conference
> on Machine Learning_, 2021. \
> [10] Minyoung Kim, Da Li, Shell X Hu, and Timothy Hospedales. Fisher SAM: Information geometry
> and sharpness aware minimisation. In _International Conference on Machine Learning_, 2022.

---

> ### Author Response · Authors · 2022-11-17
> **Reply to Reviewer qUYU (1/2)**
>
> Thank you for your thoughtful review and valuable feedback. We address your concerns as follows.
>
> ---
>
> > Q1.
> "choosing this threshold is not well justified.
> Assuming that the stochastic gradient follows a normal
> distribution is not well justified.
> For example, there exists research in the literature that
> shows this norm follows heavy-tailed distribution
> and not normal distribution."
>
> **A1.**
> Currently, there are still different opinions
> on the distribution of stochastic gradient.
> [1,2] use a heavy-tailed
> distribution (Lévy distribution),
> while [3-5] use a normal distribution.
> In our paper,
> we observed the distribution of stochastic gradient norm follows a
> Bell curve (Figure 3(a) in the paper,
> and the newly added Figures 18 and 19 in Appendix B.3),
> and thus we use the normal distribution.
> Note also that this normality assumption is not
> required in the theoretical analysis.
>
> ---
>
> > Q2.
> "In your analysis,
> the adaptivity of the threshold doesn’t show up and the analysis is very similar to the method which
> takes SAM step randomly. "
>
> **A2.**
> Thanks for pointing this out. We have clarified the analysis in Section 3.3
> accordingly.
> Indeed,
> the convergence analysis
> in Theorem 3.3
> holds for any algorithm that mixes SAM and ERM (and thus includes
> SS-SAM [6] and SAM-$k$ [7]).
> The convergence rate depends on the fraction of SAM updates
> $\zeta=\frac{1}{T} \sum_{t=0}^{T-1} \xi_t$,
> where $\xi_t\in \{0,1\}$ indicates whether SAM or ERM is used at iteration $t$
> (i.e., $\xi_t=1$ for SAM, and $0$ for ERM).
> In the proposed adaptive policy (Step 6 of Algorithm 1),
> $\xi_t = \mathbb{I}_{\\{{\bf w}: \Vert\nabla \mathcal{L}(\mathcal{B}_t; {\bf w}) \Vert^2 \geq \mu_t + c_t\sigma_t\\}}({\bf w}_t)$, which
> depends on the threshold
> $\mu_t + c_t\sigma_t$.
> However, in SS-SAM,
> $\xi_t$ is
> simply sampled from a Bernoulli distribution, and
> thus not adaptive to the loss landscape.
> Moreover, the theorem can also be used on SAM, and is shown in Corollary A.2 of Appendix A.1.
>
> ---
>
> >Q3.
> "Specifically, in the deterministic setting,
> there is no stochastic gradient and hence normal
> distribution assumption is incorrect. "
>
>
> **A3.**
> As mentioned in the above reply,
> we only consider the dependence of the convergence rate on the fraction of SAM
> updates.
> Hence, it can also be used in the deterministic setting. However,
> since typical deep networks are trained in a stochastic manner,
> convergence result in the deterministic setting is less important, thus we have moved it to
> Appendix
> A.2 in this revised version.
>
> ---
>
> > Q4. "in the empirical result, there is no significant
> difference between AE-SAM and
> SS-SAM."
>
> **A4.**
> For the *CIFAR* results in
> Table 1,  the accuracy of
> SS-SAM is already very high, and so
> a big improvement over
> SS-SAM
> is difficult.
> However, for datasets with label noise,
> Table 3
> shows that
> AE-SAM outperforms SS-SAM by a **large** margin.

---

### Author Response · Authors · 2022-11-17
**Reply to all the reviewers**

Dear Reviewers,

We thank all the reviewers for their constructive and valuable comments.  We have revised the paper accordingly,
please check the updated version.  Major changes are highlighted in blue.

We have responded to each reviewer separately.
We hope that we have satisfactorily addressed
your concerns.

Please let us know if there are any further concerns or questions.

Best,

The authors.

---

### Decision · Program_Chairs · 2023-01-20

**Decision:**

Accept: poster

**Justification For Why Not Higher Score:**

While the paper has new and interesting ideas with promising results, the ideas are not deep enough to be considered for a spotlight or oral.

**Justification For Why Not Lower Score:**

The scores are overall on the accept side, the authors did a great job with rebuttal (which resulted in increase of score by reviewers).

**Metareview: Summary, Strengths And Weaknesses:**

This paper presents a framework for reducing the computational cost of SAM algorithm. The latter is state-of-the-art optimizer, but the computational cost of it is almost twice as that of SGD. The paper proposes a simple and interesting strategy for dynamically switching between SGD and SAM through the optimization trajectory, and provides a theoretical convergence guarantee for it. The rationale behind the idea is that SAM is not much useful at places that the landscape is already flat. In such regimes, one can apply SGD to navigate in the loss landscape. However, SAM gets critical in regions when the landscape is sharp, where it is needed to take the trajectory in directions that reduces this sharpness. The notation of sharpness used by the paper is based on norm of the gradient, which is a sound proxy for measuring sharpness and can be related to the notion of sharpness used by SAM. However, since the estimate of gradient norm is computed from minibatches, the estimates are noisy. In order to reduce this noise, the paper refines the estimate by exponential moving average.

In my opinion, the paper is very well-written, has a good balance between theory and experiments, presents a sound, interesting and effective idea, and has a great coverage of related works. Some concerns were raised by the reviewers, but the authors did a great job in handling those questions, as a result of which, the reviewers increased their score. Among these was an issue related to the convergence proof (not considering adaptivity of the threshold value), which authors clarified and modified their proof in Section 3.3 to address that. Overall, the paper is borderline accept, but I think it has interesting contributions to be an accept. Please make sure the final version of the paper addresses the clarity issues raised by the reviewers.


**Note From Pc:**

if the above contains the word "oral" or "spotlight" please see: "oral" presentation means -> notable-top-5% and "spotlight" means -> notable-top-25%. As stated in our emails, we are disassociating presentation type from AC recommendations

**Summary Of Ac-Reviewer Meeting:**

As a result of this, two reviewers increased their score. In particular, reviewer qUYU had some concern about the proof of convergence in the paper, but apparently the reviewer had not looked at the revised version of the paper (Section 3.3), where corrected proof was added. After discussing about this correction, the reviewer commented posted that the idea of the paper is now clear and he increased the score. Overall, although the ratings add up to borderline accept, I find authors response quite strong and I find the revised submission in a good shape.